# CASB: a concanavalin A-based sample barcoding strategy for single-cell sequencing

Liang Fang[1,2,3,*,†] (ID), Guipeng Li[1,2,3,†] (ID), Zhiyuan Sun[2], Qionghua Zhu[1,2], Huanhuan Cui[1,2,3], Yunfei Li[2], Jingwen Zhang[2,3], Weizheng Liang[2] (ID), Wencheng Wei[2], Yuhui Hu[1,2] (ID) & Wei Chen[1,2,3,**] (ID)

## Abstract

**Sample multiplexing facilitates single-cell sequencing by reducing costs, revealing subtle difference between similar samples, and identifying artifacts such as cell doublets. However, universal and cost-effective strategies are rather limited. Here, we reported a concanavalin A-based sample barcoding strategy (CASB), which could be followed by both single-cell mRNA and ATAC (assay for transposase-accessible chromatin) sequencing techniques. The method involves minimal sample processing, thereby preserving intact transcriptomic or epigenomic patterns. We demonstrated its high labeling efficiency, high accuracy in assigning cells/nuclei to samples regardless of cell type and genetic background, and high sensitivity in detecting doublets by three applications: 1) CASB followed by scRNA-seq to track the transcriptomic dynamics of a cancer cell line perturbed by multiple drugs, which revealed compound-specific heterogeneous response; 2) CASB together with both snATAC-seq and scRNA-seq to illustrate the IFN-γ-mediated dynamic changes on epigenome and transcriptome profile, which identified the transcription factor underlying heterogeneous IFN-γ response; and 3) combinatorial indexing by CASB, which demonstrated its high scalability.**

**Keywords** CASB; combinatorial sample indexing; sample multiplexing; single-cell RNA sequencing; single-nucleus ATAC sequencing

**Subject Categories** Chromatin, Transcription & Genomics; Methods & Resources

**Mol Syst Biol. (2021) 17: e10060**

## Introduction

Single-cell mRNA sequencing (scRNA-seq) and single-nucleus assay for transposase-accessible chromatin using sequencing (snATAC-seq) have emerged as powerful technologies for interrogating the heterogeneous transcriptional profiles and chromatin landscapes of multicellular subjects (Hashimshony *et al*, 2012; Ramskold *et al*, 2012; Buenrostro *et al*, 2015; Cusanovich *et al*, 2015). Early scRNA/snATAC-seq workflows were limited to analyzing tens to hundreds of individual cells at a time. With the latest development of single-cell sequencing technologies based on microwells (Gierahn *et al*, 2017), combinatorial indexing (Cusanovich *et al*, 2015; Cao *et al*, 2017; Cao *et al*, 2018; Rosenberg *et al*, 2018; Cao *et al*, 2019) and droplet-microfluidics (Klein *et al*, 2015; Macosko *et al*, 2015), the parallel analysis of thousands of single cells or nuclei has become routine. The increase in throughput does not only lower the reagent costs per cell, but also enable the analysis of whole organs or entire organisms in one experimental run.

Recently, with the ever-increasing throughput, these technologies have also been used to reveal the temporal response of heterogeneous cell population under diverse perturbations, which require tens of samples to be processed in parallel (Hurley *et al*, 2020; Weinreb *et al*, 2020). Based on existing methods, sample-specific barcodes (for example, Illumina library indices) are often incorporated at the very end of standard library preparation workflow. Such workflow requires parallel processing of multiple individual samples until the final step, therefore not only is labor-intensive and limits the number of samples, but also increase the reagent costs if a small number of cells would be sufficient to characterize the heterogeneity of each individual sample. To overcome this, alternative multiplexing approaches should label cells from each sample with distinct barcodes before pooling for single-cell sequencing experiment. The sample-specific barcodes could then be linked to cell barcodes during single-cell sequencing library preparation. Several methods have been developed in this endeavor, which introduce sample barcodes using either genetic or non-genetic mechanisms. Genetically, researchers have used various strategies to express an exogenous gene with sample-specific barcodes at its 3′ UTR, which can be captured similarly as endogenous genes (Hurley *et al*, 2020; Weinreb *et al*, 2020); non-genetically (summarized in Table EV1), people have used oligonucleotide containing a sample barcode followed by a poly-A sequences, which can be immobilized on the cell or nuclear membrane through anchoring molecules (e.g., antibody and lipid) (Stoeckius *et al*, 2017; McGinnis *et al*, 2019) or chemical

1 Shenzhen Key Laboratory of Gene Regulation and Systems Biology, School of Life Sciences, Southern University of Science and Technology, Shenzhen, China
2 Department of Biology, School of Life Sciences, Southern University of Science and Technology, Shenzhen, China
3 Academy for Advanced Interdisciplinary Studies, Southern University of Science and Technology, Shenzhen, China
*Corresponding author. Tel: +86 138 2328 8350; E-mail: fangl@sustech.edu.cn
**Corresponding author. Tel: +86 755 8801 8449; E-mail: chenw@sustech.edu.cn
†These authors contributed equally to this work

cross-linking reaction (Gehring *et al*, 2020), or defused into permeabilized nuclei (Srivatsan *et al*, 2020), and then captured during reverse transcription. Although being already quite powerful, each of these methods has still its own limitations, including issues with scalability, universality, or the potential to introduce artifactual perturbations. Moreover, all of these methods have only been combined with scRNA/snRNA-seq and have not yet been applied and are likely incompatible with snATAC-seq.

Here, we developed a concanavalin A-based sample barcoding strategy (CASB) that overcomes many of these limitations. Taking advantage of the glycoprotein-binding ability of concanavalin A (ConA), CASB was used to label cell or nucleus with biotinylated single-strand DNA (ssDNA) through a streptavidin bridge. CASB could be easily adapted into scRNA/snATAC-seq workflows and showed high accuracy in assigning cells or nuclei regardless of genetic background as well as in resolving cell doublets. The application of CASB in samples with time-series experiments, followed by scRNA- and/or snATAC-seq, allows revealing diverse transcriptome/epigenome dynamics.

# Results

### CASB enables cell and nucleus labeling with ssDNA

The CASB complex consists of three components: biotinylated ConA, streptavidin, and biotinylated ssDNA as barcoding molecules. Both ConA and streptavidin form homo-tetramer autonomously, allowing the assembly of ConA-streptavidin-ssDNA complex (Fig 1A). Relying on the glycoprotein-binding ability of ConA, such assembled complex can be immobilized on the cell or nuclear membrane (Fig 1A). To measure how many ssDNA molecules can be immobilized on the cell membrane, a biotinylated ssDNA with 5′ and 3′ PCR handles flanking an eight-nucleotide (N8) random sequence was used to label the cells (Fig 1A). After incubation with different quantity of preassembled ConA-streptavidin-ssDNA complex in DPBS on ice (Methods), the number of ssDNA molecules immobilized on mouse embryonic stem cells (mESC) was quantified using qPCR. As shown in Fig 1B, the amount of ssDNA immobilized on cells increased with the increased usage of ConA-streptavidin-ssDNA complex and could reach as many as 50,000 molecules per cell. To test whether ssDNA may fall off from labeled cells and cause cross-contamination during sample pooling, a mouse embryonic fibroblast (MEF) cell population expressing mCherry fluorescent proteins was labeled with the ssDNA and then mixed with another MEF cell population expressing GFP fluorescent proteins, which was only coated with "empty" ConA (Methods). After 30 min incubation in DPBS on ice, mCherry and GFP positive cells were separated using fluorescence-activated cell sorting (FACS) and subjected to qPCR measurement. As shown in Fig 1C, the ssDNA immobilized on mCherry$^+$ cells were not detectable from GFP$^+$ cells, demonstrating the stability of CASB labeling. In addition to labeling the whole cell, we also measured the labeling efficiency of CASB for cell nucleus, in which nuclei were labeled with preassembled ConA-streptavidin-ssDNA complex in nuclear extraction buffer on ice (Methods). As shown in Fig EV1A, the amount of ssDNA immobilized on nuclei increased with the increased usage of CASB complex and reached at least 120,000 molecules per nucleus. Given that cell

or nucleus aggregation could significantly affect single-cell sequencing experiments, we examined whether CASB may cause cells or nuclei to aggregate. Both imaging and flow cytometry analysis demonstrated that ConA-streptavidin-ssDNA complex did not induce cell or nucleus aggregation (Fig EV1B and C). Taken together, these results demonstrated that CASB is able to stably label both cell and nucleus with biotinylated ssDNA and potentially suitable for single-cell sequencing experiments.

### CASB enables scRNA-seq sample multiplexing

In scRNA-seq, cell-specific barcodes are attached to the cDNA during reverse transcription (RT) by using primers consisting of a cell barcode sequence, a unique molecular identifier (UMI) sequence, and a poly-T sequence that anchors to the poly-A tail of mRNA molecule. To make our CASB compatible with the standard scRNA-seq workflow, we designed a biotinylated barcoding ssDNA with a 5′ PCR handle followed by a N8 barcode and a 30 nt poly-A tail, which can be captured by a RT primer consisting of a PCR handle followed by a 30 nt poly-T tail (Fig 2A). After CASB labeling, MEF cells were directly lysed and subjected to RT reaction (Methods). The barcoding ssDNA immobilized on cell membrane was quantified together with the endogenous housekeeping gene ActB using qPCR. As shown in Fig EV2A, both barcoding ssDNA and ActB gene can be efficiently captured by RT primer. Therefore, CASB could be easily adapted into scRNA-seq workflow with high efficiency.

To demonstrate the strength of CASB in scRNA-seq, a breast cancer cell line MDA-MB-231 was perturbed with 5 different compounds, collected at 3 different time points after treatment, and pooled with 3 other breast cancer cell lines as well as MEF cells after separate sample labeling using CASB (Fig 2B). Unlabeled MDA-MB-231 cells were also added into the sample pool to measure the potential influence of CASB on transcriptome profile. Sample pool was then subjected to scRNA-seq using the 10× Genomics Chromium system with minor modifications: (i) in order to examine the efficiency of CASB to detect doublets, we intentionally overloaded the system (~20,000 instead of ~10,000 cells recommended by the manufacturer) to create more cell doublets; (ii) CASB barcode and transcriptome library were separated by size selection before next-generation sequencing library construction, enabling pooled sequencing at user-defined proportions (Methods).

As a result, a total of 12,068 cells were captured with sufficient reads for transcriptome analysis. For each cell, the reads derived from each of the 20 different sample barcodes were counted and used to demultiplex the samples using HTODemux method (Stoeckius *et al*, 2018) (Methods). A total of 483 cells were assigned as "Unlabeled", as expected due to the inclusion of unlabeled MDA-MB-231 cells (Fig EV2B). Among the remaining ones, 3,962 cells were assigned as cell doublets encapsulated in the same droplet, as they contained two or more major barcodes (Fig EV2B). Indeed, the doublets consisting of both mouse and human cells, which could be unambiguously detected based on their mapping results, could also be efficiently identified based on the mixture of CASB barcodes. As shown in Fig 2C, out of 110 mouse-human doublets, 107 (97.3%) were defined as doublets based on our CASB data. When compared with singlets, more UMI derived from both CASB barcode and mRNA transcripts were detected in doublets (Fig EV2C), further validating the correct assignment of cell doublets. Within 7,623

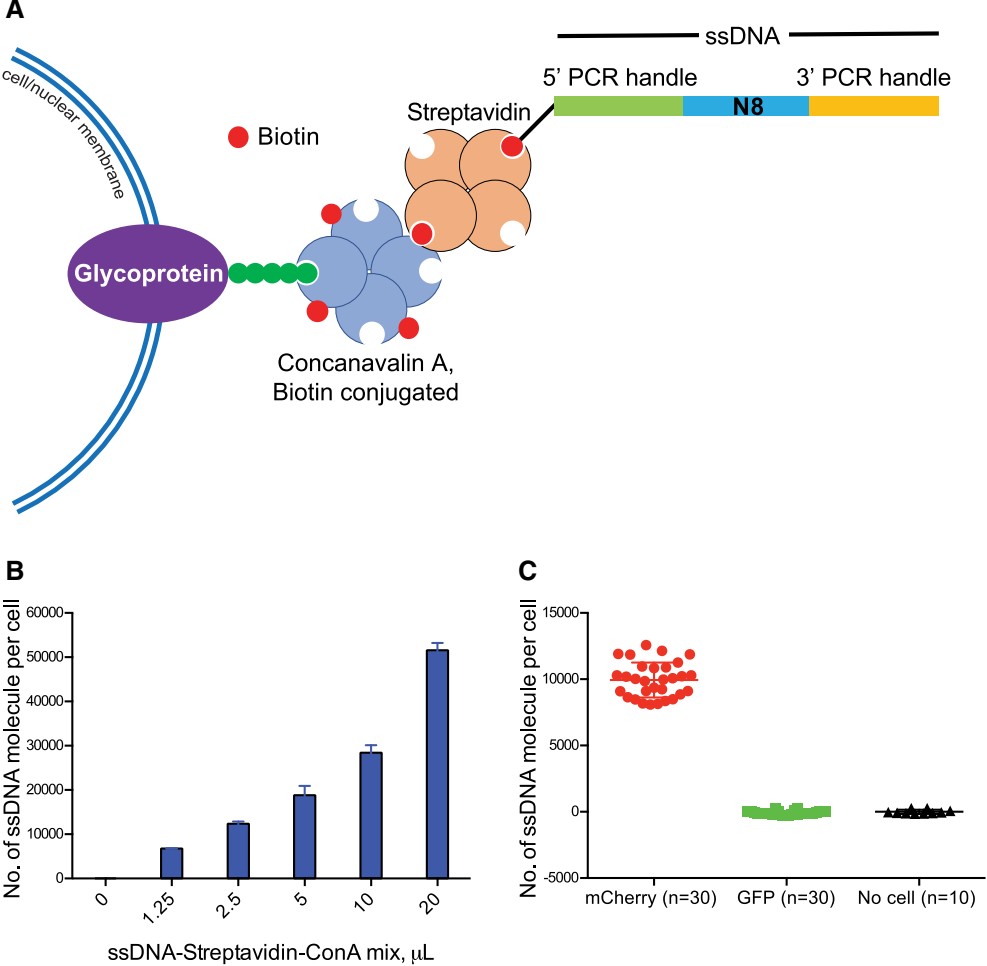

**Figure 1. Cell labeling with CASB.**

A An illustration of CASB. Biotinylated ssDNA was immobilized on glycoprotein on cell/nuclear membrane through streptavidin and biotinylated ConA. The ssDNA contains 5′ and 3′ PCR handles that flank an 8 nt random sequence.

B mESC were labeled with different quantity of CASB, and the number of ssDNA molecules immobilized on mESC was quantified used qPCR. The amount of ssDNA immobilized on cells increased with the increased usage of ConA-streptavidin-ssDNA complex and reach as many as 50,000 molecules per cell. Three independent biological replicates were performed. Error bars represent SD.

C CASB-labeled mCherry[+] MEF cells were incubated with unlabeled GFP[+] MEF cells. The number of ssDNA molecules immobilized on mCherry[+] and GFP[+] cells was quantified used qPCR after FACS separation. The ssDNA immobilized on mCherry[+] cells was not detectable from GFP[+] cells. "n" means number of qPCR reactions. Error bars represent SD.

singlets, the number of detected UMI from CASB per cell ranged from 245 to 2,134 (5–95 percentile) and significantly correlated with UMI detected for endogenous transcripts in the same cell (Fig EV2D), suggesting a similar cell-specific capture efficiency between CASB barcode and endogenous transcripts, and that CASB did not impair mRNA capture. Based on the qPCR quantification, that the same amount of CASB mixture could label cells with ~20,000 ssDNA (Fig 1B), mean UMI (1,051) detected in the scRNA-seq indicates a ~5% capture efficiency at current sequencing depth (25 million total sequencing reads). To determine the variation of labeling efficiency among different cells, given the cell-specific capture efficiency, we first normalized the number of UMI numbers from CASB by that of UMI derived from the endogenous transcripts in the same cell. As shown in Fig EV2E, the CASB barcoding

manifested a good uniformity of labeling efficiency among all singlets and within individual cell samples. Taken together, our CASB strategy could achieve high sensitivity in detecting cell identity and doublets in scRNA-seq experiments.

For the 7,623 cells with unambiguously assigned sample origin, we then clustered them based on their scRNA-seq profiles. As shown in Fig 2D, different human and mouse cells formed 5 distinct cell clusters, respectively. Each cluster was composed of cells from individual cell line labeled with distinct CASB indices (Fig 2D). We also compared the untreated MDA-MB-231 cells with to those without CASB labeling. As shown in Fig EV2F and G, single-cell profiles were intermingled together and their cumulative transcriptome was highly correlated, demonstrating a negligible influence of CASB labeling on transcriptome profile. Within the MDA-MB-231 cell

population, all 16 sample barcodes can be detected (Fig EV2H). Cells associated with 24 h-treatment of Niraparib, Rucaparib, and OSI-027 could be well distinguished from untreated cells, whereas those with LCL161 and Fludarabine could not (Fig EV2I). As

expected, Niraparib- and Rucaparib-treated cells were intermingled due to their common molecular target PARP.

MDA-MB-231 is of triple-negative breast cancer origin, which lacks efficient targeted therapy. As intratumoral heterogeneity has

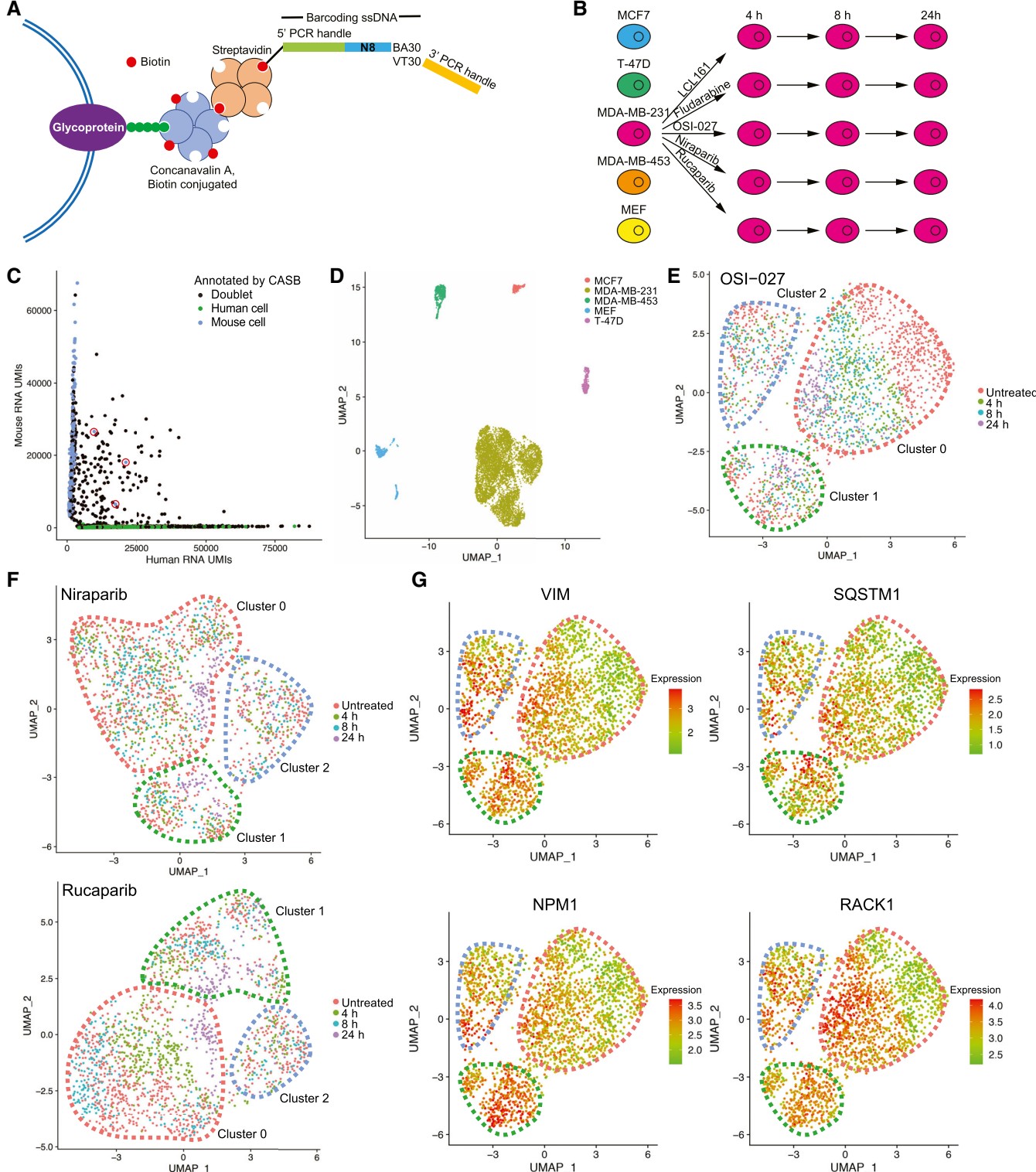

**Figure 2.**

◀

**Figure 2.   CASB enables scRNA-seq sample multiplexing.**

A   An illustration of CASB used in scRNA-seq. A biotinylated barcoding ssDNA with a 5′ PCR handle followed by an 8 nt barcode and a 30 nt poly-A tail was used to mimic the endogenous transcripts.

B   The design of the experiment. MDA-MB-231 cells were perturbed with 5 different compounds, collected at 3 different time points, CASB-labeled, and then pooled with 3 other breast cancer cell lines and MEF cells.

C   Scatter plot depicting the number of UMIs associated with transcripts from human or mouse genome. Cell doublets revealed by CASB were marked in black. Out of 110 mouse-human doublets, 107 were detected as doublets by CASB barcodes. Three interspecies cell doublets that were not detected by CASB were circled in red. Beside interspecies cell doublets, cell doublets from one species were also detected by CASB.

D   Transcriptome-based UMAP of cells captured in scRNA-seq. Cells were colored according to the CASB barcodes, and doublets were excluded. Different human and mouse cells formed 5 distinct cell clusters, respectively.

E, F   Transcriptome-based UMAP of untreated and (E) OSI-027-, (F) Niraparib- and Rucaparib-treated MDA-MB-231 cells. Three cell populations with distinct transcriptomic responses were observed in each UMAP: Sensitive cell subpopulation was circled in red, while insensitive ones in green and blue, respectively.

G   Transcriptome-based UMAP of untreated and OSI-027-treated MDA-MB-231 cells. Sensitive cell subpopulation was circled in red, while insensitive ones in green and blue, respectively. Relative expression level of VIM, SQSTM1, NPM1, and RACK1 is indicated by color code, which was expressed in untreated insensitive cell populations and induced by OSI-027 in sensitive cells.

been associated with therapy resistance, we investigated whether drug treatments could lead to heterogeneous response in the MDA-MB-231 cells. We focused on compound OSI-027, as it induced the largest transcriptomic changes (Fig EV2I). As shown in Figs 2E and EV3A and B, in which cells treated with OSI-027 were plotted with untreated cells, there indeed existed three cell populations with distinct transcriptomic responses. Although one showed clearly time-dependent transcriptomic changes (Fig 2E, cluster 0 circled in pink), the other two had limited alteration in gene expression even after 24 h (Fig 2E, cluster 1 and 2 circled in green and blue, respectively), suggesting that the latter were less sensitive to the OSI-027. Neighbor proportion analysis also confirmed that untreated cells were well separated from treated cells in cluster 0, while it is not the case for cluster 1 and 2 (Fig EV3C). As demonstrated in Fig EV3D, three distinct cell clusters could already be observed in the untreated MDA-MB-231 cells, in which specific marker genes could be identified (Fig EV4A). Next, we sought to further check whether the insensitive cell populations were also resistant to other two effective compounds, Niraparib and Rucaparib, suggested in Fig EV2I. Indeed, as shown in Figs 2F and EV3E, cells in cluster 1 and 2 also appeared less sensitive to Niraparib and Rucaparib, suggesting the intrinsic multidrug insensitivity. Differential gene expression analysis revealed that OSI-027, Niraparib, and Rucaparib induced expression alteration of 613, 365, and 296 genes (|logFC| > 0.25, *P*-value < 0.05, Dataset EV1) in the sensitive cell population (cluster 0), respectively, which are highly enriched in cell death and survival pathway (Fig EV3F), suggesting their potency on this cell population.

To explore the underlying factors of drug insensitivity, we perform function enrichment analysis on genes that were commonly up- or downregulated in cluster 1 and 2 compared to cluster 0 using IPA software (Methods). Interestingly, these genes were highly enriched in the cellular compromise and movement pathways (Dataset EV2 and Fig EV3G). Importantly, many genes upregulated in cluster 1 and 2 are known to promote cellular movement (Fig EV3H). In tumor cells, increased cell motility mediated by epithelial–mesenchymal transition (EMT) is highly associated with drug resistance (Singh & Settleman, 2010; Zhang & Weinberg, 2018). Our results suggested that the intrinsic multidrug insensitivity of MDA-MB-231 cells may result from the activated EMT. More interestingly, while overlapping the potential insensitivity-causing genes in cluster 1 and 2 with OSI-27-regulated genes in cluster 0, we observed that many genes, including VIM, SQSTM1, NPM1, and

RACK1, were induced by OSI-027 in sensitive cells (Fig 2G). Given these genes are involved in promoting EMT and potentially also drug resistance (Fig EV3F), this observation indicated the potential of OSI-027 treatment in inducing acquired therapy resistance.

**CASB enables snATAC-seq sample multiplexing**

So far, no sample multiplexing method has been developed for droplet-based snATAC-seq. In droplet-based snATAC-seq, cell nuclei are firstly incubated with transposase in bulk, where genomic DNA is fragmented and tagged with adapter sequences (also referred to as "tagmentation"). Afterward, single nuclei are encapsulated, and cell barcodes are added to DNA fragments during PCR in individual droplets using primers targeting the adapter sequences. To adapt CASB into snATAC-seq workflow, we designed a 222 nt barcoding ssDNA with S5-ME and S7-ME adapter sequences flanking a sequence containing sample barcodes (Fig 3A). S5-ME and S7-ME adapter sequences were used as primer anchoring sites during snATAC-seq library amplification (Methods). The labeling efficiency using such ssDNA was measured similarly as before, in which nuclei were directly labeled with preassembled ConA-streptavidin-ssDNA complex in nuclear extraction buffer on ice (Methods). As shown in Fig EV5A, the amount of ssDNA immobilized on nuclei increased with the increased usage of ConA-streptavidin-ssDNA complex and could reach at least 80,000 molecules per nucleus.

Before applying CASB for large-scale snATAC-seq, we tested whether CASB is compatible with tagmentation reaction and may interfere with epigenomic profile using plate-based snATAC-seq. In this experiment, CASB-labeled HAP1 cells were pooled with the same number of unlabeled HAP1 cells and then subjected for bulk tagmentation, FACS, and ATAC-seq library preparation (Methods). In total, 162 out of 192 cells, which were collected by FACS in two 96-well plates, were obtained with sufficient quality. We then separate CASB-labeled HAP1 cells from unlabeled cells based on the number of CASB barcode reads in individual cells (Fig EV5B). As shown in Fig EV5C, CASB-labeled and unlabeled cells were intermingled in UMAP projection according to the ATAC signal, suggesting no influence of CASB labeling on epigenomic profile. This was further confirmed, when the cumulative ATAC signal of the labeled and unlabeled cells was compared: The correlation between the labeled and unlabeled cells was similar as that between the two plates (Fig EV5D).

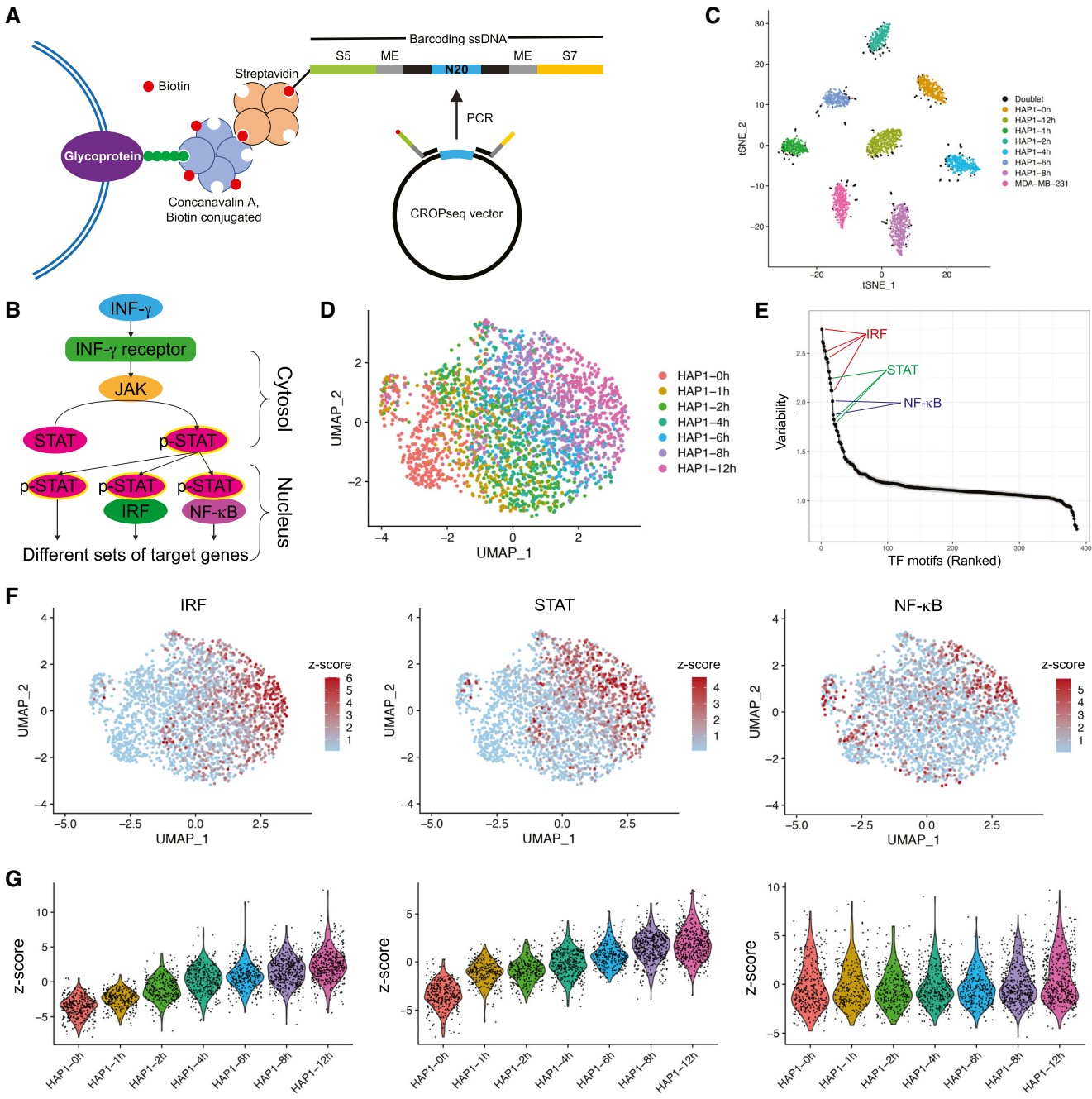

**Figure 3. CASB enables snATAC-seq sample multiplexing.**

A   An illustration of CASB used in snATAC-seq. A biotinylated barcoding ssDNA with S5-ME and S7-ME adapter sequences flanking a sequence containing sample barcodes was used to mimic the transposed genomic DNA.

B   A simplified illustration of INF-γ signaling pathway. Upon binding of INF-γ to its receptor, JAK is activated and induces the phosphorylation of STAT. Phosphorylated STAT is then translocated into the nucleus and activates the expression of different sets of target genes by itself or in combination with other transcription factors.

C   t-SNE projection based on the CASB barcode reads captured in snATAC-seq. Cells were colored according to the CASB barcodes, and doublets were marked in black.

D   ATAC-based UMAP of all HAP1 cells captured in snATAC-seq. Cells were colored according to the CASB barcodes, and doublets were excluded. HAP1 cells showed a continuous shift in chromatin profile from 0 to 12 h.

E   Dot plot revealing the TFs with the most variable activity across all cells including IRF, STAT and NF-κB.

F   ATAC-based UMAP of all HAP1 cells, in which the TF activity was presented by bias-corrected deviation z-score across all cells in color code.

G   Violin plots demonstrating the deviation z-score of different TFs across different cells at different time points. Each dot represents a cell. While IRF and STAT activity showed continuous upregulation upon IFN-γ stimulation, the activity of NF-κB remained unchanged but showed high heterogeneity within HAP1 cells.

To demonstrate the application of CASB in large-scale snATAC-seq, we sought to monitor the temporal chromatin changes induced by interferon-gamma (IFN-γ) in HAP1 cells. IFN-γ is an important cytokine in the host defense against infection by viral and microbial pathogens (Shtrichman & Samuel, 2001). It mediates innate immunity through regulating effector gene expression (Fig 3B), which is accompanied by substantial changes at epigenetic level (Ivashkiv, 2018). However, how heterogeneously and dynamically cells respond to IFN-γ stimulation at the chromatin level has remained elusive. Taking advantage of CASB, we analyzed the changes in chromatin accessibility of HAP1 cells at 7 different time points after IFN-γ stimulation using snATAC-seq. MDA-MB-231 cells were added into the pool here as an outlier control. Of a total of 345,538,181 sequencing reads, the 5,095,947 (about 1.5%) were derived from CASB barcodes. 3,218 cells were obtained with sufficient reads, 305 of which were identified as cell doublets that have at least two major CASB barcodes (Fig 3C, marked in black), and 23 cells were unlabeled (Fig EV5E and F). MDA-MB-231 cells with its specific CASB barcode presented as an isolated cluster (Fig EV5G). As revealed by UMAP projection, HAP1 cells showed a continuous shift in chromatin profile from 0 to 12 h (Fig 3D). To uncover the key transcription factors (TFs) that mediate IFN-γ-induced chromatin remodeling, we analyzed their binding motifs on the ATAC peaks across all cells and observed that peaks containing motifs of IRF, STAT, and NF-κB TF showed large variation in their intensity, indicating their critical functions in modulating IFN-γ response (Fig 3E) (Methods). Indeed, IRF- and STAT-associated peaks showed continuous activation across different time points (Fig 3F and G, left and middle panel). This is expected as IFN-γ activates JAK/STAT signaling through binding to its receptor, which in turn activates the expression of IFN-γ-responsive genes through transcription factors STATs and IRFs (Fig 3B) (Leonard & O'Shea, 1998). Interestingly, the large variation of NF-κB peak intensity did not result from IFN-γ treatment but was instead largely due to the heterogeneity within HAP1 cells (Fig 3F and G, right panel). It is known that NF-κB can be activated by IFN-γ and is able to facilitate the transcription activation of IFN-γ targets, including CXCLs (Qin *et al*, 2007; Pfeffer, 2011). This result suggested that heterogeneous NF-κB activity may give rise to heterogeneous IFN-γ response.

To evaluate whether heterogeneous NF-κB activity causes heterogeneous IFN-γ response, IFN-γ treated samples from the same time points were also analyzed using CASB followed by scRNA-seq. A total of 3,407 cells were captured, 294 of which were identified as cell doublets and 9 cells were unlabeled (Fig EV6A and B). As shown in Fig 4A, HAP1 cells showed a continuous shift in the transcriptome profile from 0 to 12 h on UMAP projection. To globally evaluate the correlation between chromatin accessibility and gene expression, we analyzed the dynamic expression patterns of predicted IRF and STAT target genes. In consistent with the activity of IRF and STAT observed in snATAC-seq (Fig 3G, left and middle panel), the expression of their target genes also exhibited continuous upregulation (Fig 4B).

As revealed by unsupervised clustering with Louvain method, cells at later time points (4, 6, 8, and 12 h) were clustered into two populations, one of which exhibited more divergent transcriptome profile from earlier time points (Fig 4C, cluster 2, circled in red). To see whether this is associated with heterogeneous NF-κB activity identified in snATAC-seq, the expression of predicted NF-κB target genes

was compared between the two cell populations. Consistent with the heterogeneous NF-κB activity, its target genes also exhibit heterogeneous expression and were expressed at a higher level in cluster 2 at later time points (Fig 4D). The high induction of CXCL10 and 11, well-known targets of IFN-γ, only in cluster 2 cells further corroborate that the heterogeneous NF-κB activity indeed results in differential responses to IFN-γ in HAP1 cells (Figs 4E and EV6C and D).

## CASB enables combinatorial sample indexing

Combinatorial sample indexing allows more samples to be indexed with limited number of barcodes and, therefore, to increase indexing capacity in a cost-effective way (Cusanovich *et al*, 2015; Cao *et al*, 2017; Cao *et al*, 2018; Cao *et al*, 2019). Commonly used strategies include simultaneous combinatorial indexing and sequential split-pool indexing. To test if CASB could be applied with these two strategies, we performed four-by-four combinatorial barcoding using CASB to index 16 different cell lines from seven different species (Methods), which was followed by one round of four-group split-pool barcoding to increase the complexity of barcode combinations (Fig 5A). Here, while the combinatorial barcoding will help to assign cell types and identify cell doublets consisting of cells from different cell lines (referred to as "doublets between samples"), the split-pool barcoding helps to identify cell doublets consisting of cells from the same cell line (referred to as "doublets within sample").

To investigate the influence of CASB on cell transcriptome, before loading on the 10X system for scRNA-seq, equal number of unlabeled cells from each cell type were pooled (Fig 5A). After sequencing, a total of 7,935 cells were captured with sufficient reads for transcriptome analysis, in which 4,176 cells were assigned as "Unlabeled" using HTODemux method (Methods). Based on the barcode information, we determined cell doublets between samples and within sample (Methods). Out of 3,759 labeled cells, 427 and 26 cells were assigned as "Doublet" between samples and within sample, respectively (Figs 5B and EV7A), which, as expected, showed higher UMI counts (Fig EV7B). As shown in Fig EV7C, the cell from the 16 cell lines were all labeled with sufficient amount of total CASB barcode with limited variability in labeling efficiency among different cell lines, demonstrating the universality of CASB technique.

When cell singlets were plotted on UMAP projection, distinct cell clusters could be observed, and cells from the same species were close to each other (Fig EV7D). The eight CASB barcodes used in four-by-four combinatorial labeling were exclusively distributed in individual cell clusters (Fig 5C and D), demonstrating the success of cell type-specific barcoding. Meanwhile, the four barcodes used in split-pool labeling were distributed evenly among different cell clusters (Fig 5E), showing the efficient and unbiased sequential barcoding. CASB barcode combinations successfully helped assigning cell types to the distinct cell clusters (Fig 5F and G, more detailed UMAP for human cell lines in Fig EV7D and E). Finally, to assess the influence of CASB on transcriptome profiling, labeled and unlabeled cells were compared for each of the 16 different cell lines. As shown in Fig 5H, single-cell profiles of labeled cells and unlabeled cells within distinct cell clusters were intermingled (Fig 5H), and the cumulative transcriptome profiles were also highly correlated between labeled and unlabeled cells for each of the 16 cell lines (Fig EV7F), showing a negligible influence of CASB labeling on transcriptome profile.

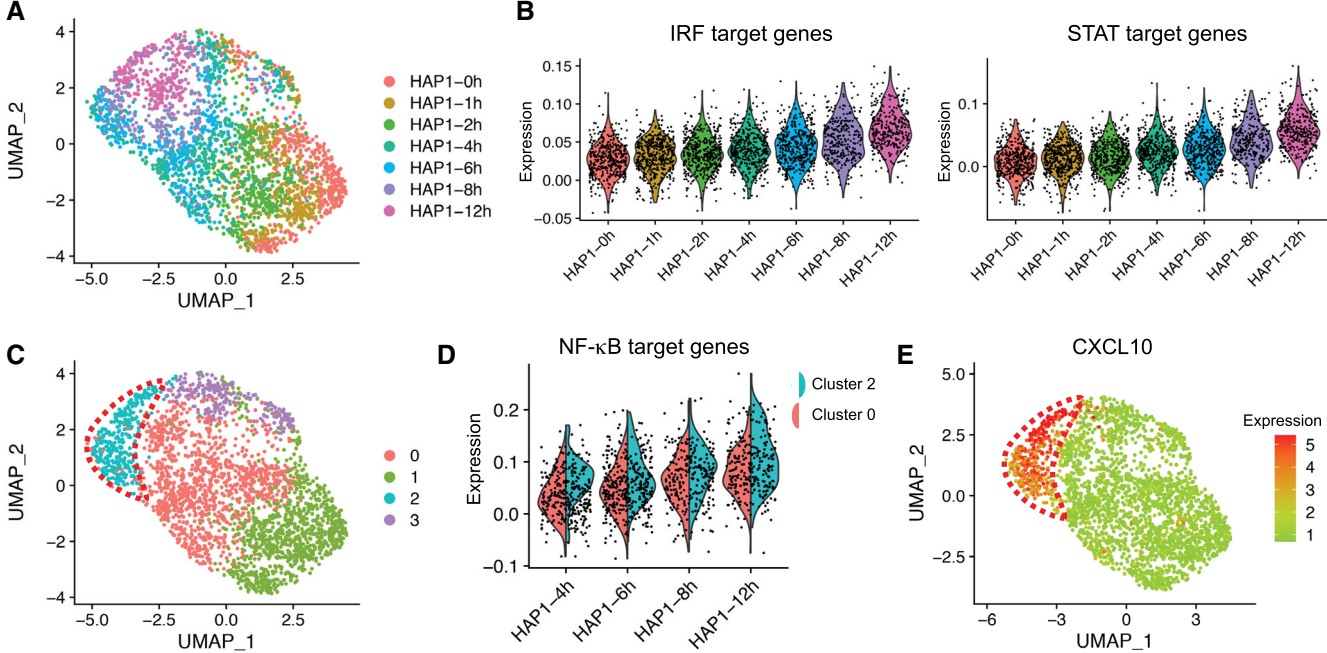

**Figure 4. Transcriptomic heterogeneity within HAP1 cells.**

A Transcriptome-based UMAP of HAP1 cells captured in scRNA-seq, in which cells were colored according to the CASB barcodes. HAP1 cells showed globally a continuous shift from 0 to 12 h.

B Violin plots demonstrating the continuous transcriptional activation of predicted IRF and STAT target genes across different time points. Each dot represents a cell, and Y-axis represents the average relative expression level of TF target genes.

C Transcriptome-based UMAP of HAP1, in which cells were unsupervised clustered and colored according to the transcriptomic feature revealed by Louvain algorithm. Cells were clustered into two populations at 4–12 h, one of which exhibited more divergent transcriptome profile from earlier time points and was highlighted with red dashed line.

D Violin plots comparing the expression of predicted NF-κB target genes between cluster 0 and 2 at 4–12 h. Each dot represents a cell, and Y-axis represents the average relative expression level of NF-κB target genes. Predicted target genes were heterogeneously expressed and more actively induced in cluster 2.

E Transcriptome-based UMAP of HAP1 cells, in which the relative expression of CXCL10 were presented with color code and showed activation only in cluster 2 (circled in red) at later time points.

Taken together, these data demonstrate both good scalability and universality of CASB method.

## Discussion

CASB is a flexible sample multiplexing approach, ready to be prepared in an average molecular biology laboratory. CASB could be used to label cells and nuclei of different cell types and from different species. Moreover, the binding of CASB molecules to the subject is fast and stable, and takes place even at low temperature, which is critical to preserve sample integrity. Importantly, the design of CASB barcoding ssDNA is extremely flexible, which can be easily adapted to different single-cell sequencing workflows.

As the first-in-class non-genetic single-cell sample barcoding technique, CITE-seq was originally developed for simultaneous quantification of cell surface epitope and transcriptome (Stoeckius *et al*, 2017). Although antibody against common epitope could be employed for multiplexing of cells from a same species (Stoeckius *et al*, 2018), CITE-seq inevitably requires different antibodies for labeling samples from different species and types (i.e., cells or nuclei). In addition, the production of conjugated antibody is

complicated and expensive. Using lipid-modified oligonucleotides that incorporate into the plasma membrane, MULTI-seq is applicable to cells or nuclei from different species (McGinnis *et al*, 2019). However, when preforming MULTI-seq, samples need to be maintained on ice to avoid the loss and exchange of labeling oligonucleotides. Given that snATAC-seq workflow includes a step of 1 h transposition reaction at 37°C, MULTI-seq is, therefore, likely incompatible with snATAC-seq. More recently, through two-step chemical reaction, ClickTags allowed methyltetrazine-modified oligonucleotides to be immobilized on the surface proteins of methanol-fixed cells (Gehring *et al*, 2020). However, its sophisticated procedure consisting of multiple washing steps may cause significant sample loss and may not be suitable for precious samples. Moreover, given ATAC-seq is not compatible with methanol-fixed sample, ClickTags is also unlikely to work for snATAC-seq. Finally, nuclear hashing strategy used in sci-Plex seems to be a straightforward method (Srivatsan *et al*, 2020), but its application is restricted to sci-Plex. Although sci-Plex achieved ultra-high throughput, it could only analyze nuclear mRNA and has significantly lower mRNA capture efficiency than droplet-based technology.

In comparison (summarized in Table EV1), CASB overcomes many above-mentioned limitations. Due to the universal presence of

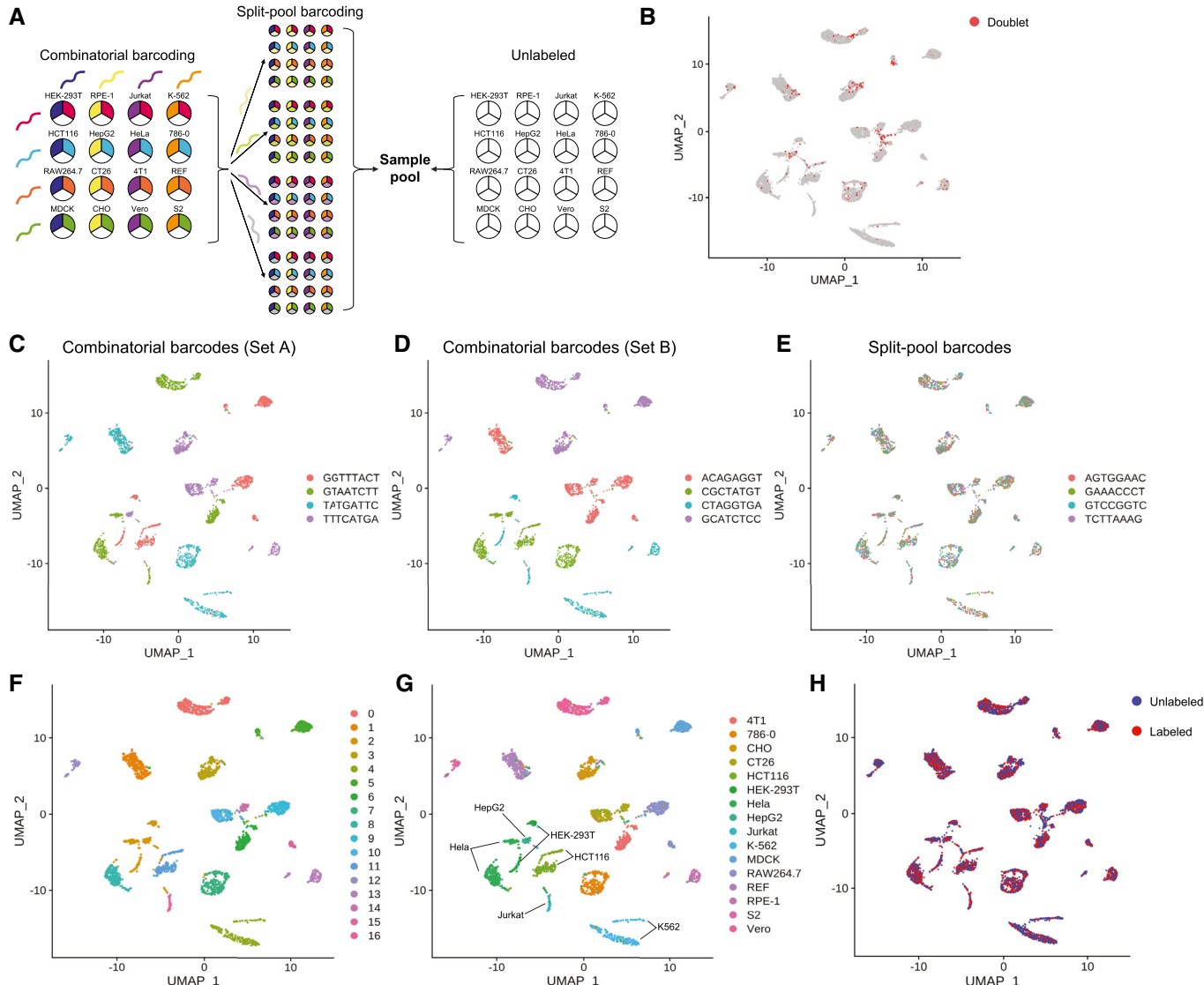

**Figure 5.  CASB enables combinatorial indexing.**

A       An illustration of the combinatorial indexing experiment. A four-by-four combinatorial barcoding strategy was used to index 16 different cell lines, which was followed by one round of four-group split-pool barcoding. The same number of unlabeled cells from the 16 cell lines was also added into the sample pool.

B–E   Transcriptome-based UMAP of 16 different cell lines, in which cells were unsupervised clustered. (B) Cell doublets were highlighted in red. (C-E) 12 CASB barcodes were indicated with different colors. Eight barcodes used in four-by-four combinatorial labeling were exclusively distributed in distinct cell clusters (C and D), while the four barcodes used in split-pool labeling distributed evenly among different cell clusters (E).

F       Cells were colored according to the transcriptomic feature revealed by Louvain algorithm.

G       Cells were colored according to CASB barcode combinations, which successfully helped assigning cell types into distinct cell clusters.

H       Labeled and unlabeled cells were marked in red and blue, respectively. They were intermingled within different cell clusters.

glycoprotein on plasma membrane, CASB is applicable to any sample with an accessible plasma membrane. Worth noting, after 1 h transposition reaction at 37°C, the CASB barcodes remained abundant and showed minimal cross-contamination. Therefore, in addition to scRNA/snATAC-seq, CASB should, in principle, be compatible with other single-cell sequencing technologies (e.g., single-cell multiome assay provided by 10X Genomics and SNARE-seq (Chen *et al*, 2019)) and work for samples preserved with different methods (e.g., flash-frozen and formalin-fixed).

CASB allows scalable sample multiplexing by solely increasing the variety of barcoding ssDNA. In this study, we tested CASB's scalability by performing a 20-plex perturbation assay followed by scRNA-seq, which revealed new information about drug response of triple-negative breast cancer cells. Specifically, it demonstrated the different response dynamics of different compounds, and different response of different cell subpopulations to the same drug. Moreover, by combining simultaneous combinatorial barcoding and sequential split-pool barcoding, we further demonstrated CASB's

high scalability. Notably, sequential barcoding has not been tested in other single-cell sample labeling techniques (summarized in Table EV1). When integrated with automated cell handling system, CASB followed by scRNA-seq could serve as a powerful platform for single-cell sequencing-based drug screens.

Cell doublets, i.e., two or more cells encapsulated in a same droplet, posed a challenge for single-cell sequencing data analysis. Without sample barcoding, cell doublets from the same species could only be estimated mathematically with certain ambiguity (DePasquale *et al*, 2019). To reduce the doublet rate, an often-sought strategy is to limit the number of cells loaded in the microw-ell- or droplet-based systems. As demonstrated in this study, CASB could reveal cell doublets in high accuracy, and as such, its applica-tion would allow to increase the throughput of single-cell sequenc-ing systems by loading more cells. Indeed, similar strategy has been proposed by using antibody-based barcoding approach (Stoeckius

*et al*, 2018). However, the efficiency of doublet identification corre-lated to the diversity of sample barcodes. By increasing the sample barcodes to hundreds or even thousands, which could be easily achieved using CASB combinatorial indexing, we would envisage a much higher doublet detection efficiency, which allows the further optimization of cell loading rate.

In summary, CASB allows to incorporate additional layers of information into single-cell sequencing experiments. With the ever-increasing throughput of single-cell sequencing technologies, CASB does not only reduce reagent costs, improve data analysis, minimize the batch effect, but also can become a versatile tool in this field by incorporating more diverse types of information, including time points, treatment conditions, and potentially also spatial coordi-nates. With further improvement, such as using ConA-Streptavidin fusion protein or fluorophore-labeled ssDNA, it will facilitate more novel applications of single-cell sequencing technology.

# Materials and Methods

## Reagents and Tools table

| Reagent/resource | Reference or source | Identifier or catalog number |
|---|---|---|
| **Experimental models** | | |
| HAP1 cells (*H. sapiens*) | Horizon discovery | C631 |
| HEK-293T cells (*H. sapiens*) | ATCC | CRL-11268 |
| MDA-MB-231 cells (*H. sapiens*) | ATCC | HTB-26 |
| MDA-MB-453 cells (*H. sapiens*) | ATCC | HTB-131 |
| T-47D cells (*H. sapiens*) | ATCC | HTB-133 |
| MCF7 cells (*H. sapiens*) | ATCC | HTB-22 |
| RPE-1 cells (*H. sapiens*) | ATCC | CRL-4000 |
| Jurkat cells (*H. sapiens*) | ATCC | TIB-152 |
| K-562 cells (*H. sapiens*) | ATCC | CCL-243 |
| HCT116 cells (*H. sapiens*) | ATCC | CCL-247 |
| HepG2 cells (*H. sapiens*) | ATCC | HB-8065 |
| HeLa cells (*H. sapiens*) | ATCC | CCL-2 |
| 786-0 cells (*H. sapiens*) | ATCC | CRL-1932 |
| RAW264.7 cells (*M. musculus*) | ATCC | TIB-71 |
| CT26 cells (*M. musculus*) | ATCC | CRL-2638 |
| 4T1 cells (*M. musculus*) | ATCC | CRL-2539 |
| mESC (*M. musculus*) | Wei Li's Laboratory | N/A |
| MEF (*M. musculus*) | Wei Li's Laboratory | N/A |
| REF cells (*R. norvegicus*) | Wei Li's Laboratory | N/A |
| MDCK cells (*C. familiaris*) | ATCC | CCL-34 |
| CHO cells (*C. griseus*) | ATCC | CCL-61 |
| Vero cells (*C. aethiops*) | ATCC | CCL-81 |
| S2 cells (*D. melanogaster*) | ATCC | CRL-1963 |
| **Recombinant DNA** | | |
| CROPseq-Guide-Puro | Addgene | #86708 |
| **Chemicals, enzymes, and other reagents** | | |
| Biotinylated ConA | Sigma-Aldrich | C2272 |

**Reagents and Tools table**  (continued)

| Reagent/resource | Reference or source | Identifier or catalog number |
|---|---|---|
| Streptavidin | Coolaber | CS10471 |
| Hieff qPCR SYBR Green Master Mix | Yeasen | 11201ES08 |
| 1st strand cDNA synthesis kit | Yeasen | 11119ES60 |
| High Sensitivity DNA Kit | Agilent | 5067-4626 |
| RMPI1640 medium | Gibco | 22400089 |
| DMEM medium | Gibco | 11995040 |
| Neurobasal medium | Gibco | 21103-049 |
| DMEM/F-12 medium | Gibco | 11330-032 |
| DPBS | Gibco | C14190500BT |
| N2 | Gibco | 17502048 |
| B27 | Gibco | 17504-044 |
| Chir99021 | Selleck | s1263 |
| PD0325901 | Selleck | s1036 |
| mLIF | Millipore | ESG1107 |
| FBS | Gibco | 10270106 |
| P/S | Gibco | 15070063 |
| IFN-γ | Peprotech | #300-02 |
| LCL161 | MedChemExpress | HY-15518 |
| Fludarabine | MedChemExpress | HY-B0069 |
| OSI-027 | MedChemExpress | HY-10423 |
| Niraparib | MedChemExpress | HY-10619 |
| Rucaparib | MedChemExpress | HY-10617 |
| Nuclei extraction buffer | Sigma-Aldrich | NUC101-1KT |
| TrionX-100 | Sigma-Aldrich | T8787 |
| Single Cell 3' Reagent Kits v2 | 10X Genomics | PN-120237 |
| Single Cell ATAC Reagent Kits | 10X Genomics | PN-1000111 |
| 2X SPRIselect Reagent | Beckman Coulter | B23318 |
| PrimeSTAR Max PCR master mix | Takara | R045A |
| TruePrep DNA Library Prep Kit V2 for Illumina | Vazyme | TD501 |
| TruePrep Index Kit V2 for Illumina | Vazyme | TD202 |
| ZYMO DNA clean & concentrator kit | ZYMO RESEARCH | D4014 |
| VAHTS DNA Clean Beads | Vazyme | N411-03 |
| Digitonin | Promega | G9441 |
| Tween-20 | Sigma-Aldrich | P7949 |
| NP40 | Sigma-Aldrich | NP40S |
| Triton X-100 | Sigma-Aldrich | T8787 |
| DAPI | Roche | 10236276001 |
| **Oligonucleotides** | | |
| Oligonucleotides | This study | Dataset EV3 |

## Methods and Protocols

### Cell culture and pre-processing

The HEK-293T, MDA-MB-231, MDA-MB-453, T-47D, MCF7, RPE-1, Jurkat, K-562, HCT116, HepG2, HeLa, 786-0, RAW264.7, CT26, 4T1, MDCK, CHO, Vero, and S2 cells were obtained from the ATCC, while HAP1 from Horizon discovery. The MEF, REF, and mESC were kindly gifted by the Wei Li's Laboratory at the Institute of Zoology, Chinese Academy of Sciences. The MDA-MB-231, MDA-MB-453, T-47D, MCF7, HEK-293T, HeLa, HCT116, HAP1, RAW264.7, CT26, 4T1, MEF, REF, MDCK, CHO, and Vero cells were cultured in DMEM medium with 10% FBS and 1% P/S with 5%

$CO_2$ at 37°C, while the mESC were cultured in Neurobasal-DMEM/F12 based medium with N2, B27, PD0325901, Chir99021 and mLIF with 5% $CO_2$ at 37°C. HAP1, Jurkat, K-562, HepG2, 786-0 cells were cultured in RMPI1640 medium with 10% FBS and 1% P/S. For stimulation with IFN-γ, HAP1 cells were treated with 100 ng/ml IFN-γ for 2, 4, 6, 8, and 12 h. For scRNA-seq-related experiments, cells were trypsinized and washed once with DPBS, while for snATAC-seq-related experiments, after washing with DPBS, cells were cryopreserved in 200 µl cryo-medium (10% DMSO, 40% FBS, 50% culture medium) and kept in −80°C.

### Compounds and treatment

The compounds used in this study include LCL161 targeting XIAP, Fludarabine inhibiting DNA synthesis, OSI-027 blocking mTOR, Rucaparib, and Niraparib targeting PARP1, which were chosen based on their selective inhibitory effect on MDA-MB-231 cells (Garnett et al, 2012; Iorio et al, 2016). Compounds LCL161, Fludarabine, OSI-027, Niraparib, and Rucaparib were obtained from MedChemExpress and dissolved in DMSO. For scRNA-seq experiment, 0.1 µM of LCL161, 0.15 µM of Fludarabine, 2.5 µM of OSI-027, 15 µM of Rucaparib, and 12.5 µM of Niraparib were used to treat the cells for 4, 8 or 24 h.

### Design and synthesis of CASB barcoding ssDNA

For measuring the number of ssDNA molecules immobilized on cell or nuclear membrane, a 5′-biotinylated ssDNA with 5′ and 3′ PCR handles flanking a N8 random sequence (Oligo-#1) was designed. For scRNA-seq-related experiments, 5′-biotinylated ssDNAs with a 5′ PCR handle followed by a N8 barcode and a 30 nt poly-A tail (Oligo-#2) were designed. For snATAC-seq-related experiments, a 5′-biotinylated forward primer (Oligo-#3) and a revers primer (Oligo-#4) were used to amplify 222 bp fragments from CROPseq-Guide-Puro plasmids which have been inserted with different gRNA sequences. To generate ssDNAs, purified PCR products were denatured at 95°C for 2 min and immediately put on ice. Information of CASB barcode sequences and their corresponding samples can be found in Dataset EV4.

### Assembly of CASB complex

Biotinylated ConA and streptavidin were dissolved in 50% glycerol at concentration of 1.6 µM and store in −20°C, while different biotinylated ssDNAs were diluted at concentration of 100 nM in nuclease-free water and stored in −20°C. To assemble CASB complex, streptavidin was firstly mixed with biotinylated ssDNA and incubated for 10 min at room temperature. Afterward, biotinylated ConA was added to the streptavidin-ssDNA mix and incubated for 10 min at room temperature. The molar ratio of streptavidin:biotinylated ssDNA:biotinylated ConA is 4:1:4.

### Cell labeling with CASB complex

1  $5 \times 10^5$ cells were collected by trypsinization and resuspended in 0.5 ml DPBS.
2  Indicated amount of assembled CASB complex was added to the cells and incubated for 10 min on ice after thorough mixing. For labeling mCherry⁺ MEF cells, 2.5 µl assembled CASB complex was used, while, for GFP⁺ MEF, 0.4 µl biotinylated ConA (1.6 µM) was used. For RT–qPCR and scRNA-seq, 5 µl assembled CASB complex was used.

3  Centrifuge cells at 300 g for 5 min at 4°C and then discard the supernatant by pipette. Resuspend cells in 0.5 ml DPBS. Repeat the centrifuge and resuspend step again.
4  Count cell number by the hemocytometer. Use appropriate number of cells for the following experiments.

### Nuclei labeling with CASB complex

1  $5 \times 10^5$ cells were thawed by adding 800 µl warm culture medium and collected by centrifugation. Afterward, cells were resuspended in 0.5 ml nuclei extraction buffer, incubated for 5 min on ice, and collected by centrifugation (500 g, 5 min, 4°C).
2  Nuclei were resuspended in 0.5 ml fresh nuclei extraction buffer containing indicated amount of assembled CASB complex. Then incubate on ice for 5 min after thorough mixing by pipette. For snATAC-seq, 2.5 µl assembled CASB complex was used.
3  Centrifuge nuclei at 500 g for 5 min at 4°C and then discard the supernatant by pipette. Resuspend nuclei in 0.5 ml nuclei wash buffer (10 mM Tris pH 7.4, 10 mM NaCl, 3 mM $MgCl_2$, 1% BSA, 0.1% Tween-20). Repeat the centrifuge and resuspend step again.
4  Count nuclei number by the hemocytometer. Use appropriate number of nuclei for the following experiments.

### Quantification of ssDNA immobilized on cell or nuclear membrane

For all quantification experiments using qPCR, standard curves were always first drawn using serially diluted pure ssDNA for calculating the precise number of ssDNA in each reaction. For each reaction of qPCR, 200 labeled cells or nuclei in 5 µl DPBS were directly mixed with 5 µl of primer mix (1 µM) and 10 µl of qPCR master mix. Primers used for quantifying barcoding ssDNA are Oligo-#5 and Oligo-#6.

For measuring the ssDNA with the poly-A tail, 20,000 cells were directly lysed in 6 µl of 0.17% TrionX-100 for 3 min at 72°C and then reverse-transcribed with first-strand cDNA synthesis kit using RT primer Oligo-#7 (final concentration 2.5 µM). The qPCR was performed with ssDNA-specific forward primer Oligo-#8 and Actb-specific forward primer Oligo-#9 combining with common reverse primer Oligo-#10.

### Aggregation analysis of cell and nucleus after CASB labeling

K562 cells were collected by centrifugation (300 g, 5 min, room temperature). $5 \times 10^5$ cells were used for cell aggregation analysis, and $5 \times 10^5$ cells were used to extract nuclei for nucleus aggregation analysis. The labeling procedures for cells and nuclei were like before. For cells and nuclei, 5 and 2.5 µl assembled CASB complex was used, respectively. After labeling, DAPI was added to nuclei suspension according to manufacturer's instruction. Afterward, cells and nuclei were imaged by fluorescence microscopy (Eclipse Ts2-FL, Nikon) and analyzed by flow cytometry (B75442, Beckman Coulter).

### Combinatorial indexing with CASB complex

1  To assemble CASB complex, 2 µl biotinylated ConA (1.6 µM), 8 µl biotinylated ssDNA (0.1 µM) and 2 µl streptavidin (1.6 µM) were used.
2  Adherent cells were collected by trypsinization, and $2.5 \times 10^5$ cells were resuspended in 0.25 ml DPBS. Suspension cells were collected by centrifugation (300 g, 5 min, room temperature), and $2.5 \times 10^5$ cells were resuspended in 0.25 ml DPBS.

3 For four-by-four combinatorial indexing, 1.25 µl of each group A ($A_1 \sim A_4$) and group B ($B_1 \sim B_4$) CASB complex were added to the cells and incubated for 10 min on ice after thorough mixing by pipette.

4 Then, centrifuge cells at 300 *g* for 5 min at 4°C and discard the supernatant by pipette. Resuspend cells in 0.25 ml DPBS. Repeat the centrifuge and resuspend step again.

5 Collect all cells into 15 ml conical centrifuge tube. Mix thorough by pipette. Split cells mixture into four 1.5 ml Eppendorf tubes (250 µl each).

6 For four-group split-pool barcoding, 0.75 µl group C ($C_1 \sim C_4$) CASB complex were added to the cells and incubated for 10 min on ice after thorough mixing by pipette.

7 Then, centrifuge cells at 300 *g* for 5 min at 4°C and discard the supernatant by pipette. Resuspend cells in 0.25 ml DPBS. Repeat the centrifuge and resuspend step again.

8 Collect all cells into a new 1.5 ml Eppendorf tube. Mix thorough by pipette.

9 Count cell number by the hemocytometer. Combine the same amount of CASB-labeled and unlabeled cells together. Use appropriate number of cells for scRNA-seq.

### scRNA-seq and snATAC-seq

The scRNA-seq experiments were performed according to the standard protocol of Single Cell 3' Reagent Kits v2 with following modifications. During cDNA amplification, additional primer (Oligo-#8, 0.1 µM) was added to amplify CASB barcode. To capture amplified CASB barcode, during "post-cDNA amplification reaction cleanup", the amplified full-length cDNA library was purified with 2× SPRIselect Reagent and eluted in 40 µl of nuclease-free water, 10 µl of which was subject to PCR with primer pair Oligo-#11 and Oligo-#12 using PrimeSTAR Max PCR master mix to add sequencing adapter sequences to CASB barcode.

The snATAC-seq experiments were performed according to the standard protocol of Single Cell ATAC Reagent Kits with no modification.

### Plate-based snATAC-seq

1 Prepare the 96-well plates by adding 1 µl 2× Lysis Buffer (100 mM NaCl pH 7.4, 100 mM Tris–HCl pH 8.0, 40 µg/ml proteinase K, 0.4 % SDS) and 1 µl of 10 µM S5xx/N7xx Nextera Index Primer Mix (5 µM each) (TD202, Vazyme) to each well.

2 Pre-coat 1.5 ml Eppendorf tubes with 500 µl 0.5% BSA/DPBS. Collect $5 \times 10^5$ cells in DPBS into 1.5 ml Eppendorf tube and then centrifuge at 300 *g* for 5 min at room temperature. Wash the cells once with 200 µl DPBS and then with 200 µl ATAC resuspension buffer (10 mM NaCl, 10 mM Tris–HCl pH 7.4, 3 mM $MgCl_2$). Centrifugation condition is 600 *g*, 5 min at 4°C.

3 After washing, resuspend cells in 100 µl ATAC cell lysis buffer (10 mM NaCl, 10 mM Tris–HCl pH 7.4, 3 mM $MgCl_2$, 0.1% Tween-20, 0.1% NP40, 0.01% Digitonin) with 2.5 µl assembled CASB complex (control group was only resuspended in ATAC cell lysis buffer). Mix thorough by pipette. Incubate on ice for 5 min, then add 950 µl ATAC wash buffer (10 mM NaCl, 10 mM Tris–HCl pH 7.4, 3 mM $MgCl_2$, 0.1% Tween-20) to quench reaction.

4 Centrifuge nuclei at 1,000 *g* for 5 min at 4°C and discard the supernatant by pipette carefully. Resuspend nuclei in 1 ml ATAC wash buffer and pool labeled and unlabeled nuclei together. Centrifuge again.

5 Resuspend nuclei in 50 µl tagmentation reaction (TD501, Vazyme). Incubate at 37°C on a ThermoMixer (600 rpm) for 30 min. Then, stop the reaction by adding 50 µl tagmentation stop buffer (20 mM EDTA, 10 mM Tris–HCl pH 8.0). Incubate on ice for 10 min.

6 Add 300 µl 0.5% BSA/DPBS into nuclei. Then, add DAPI according to manufacturer's instruction. Use FACS (MA900, Sony) to sort single nucleus into each well of prepared 96-well plates.

7 Incubate plates at 65°C for 15 min with lip temperature at 100°C.

8 Add 2 µl 10% Tween-20 into each well to quench SDS and then 6 µl PCR mix (from TD501, Vazyme). Perform PCR reaction as following: 72°C 10 min, 98°C 1 min, (98°C 30 s, 63°C 30 s, 72°C 30 s) ×25, 72°C 5 min, 10°C hold.

9 All PCR products were pooled into a 15-ml conical centrifuge tube and purified with ZYMO DNA clean & concentrator kit.

10 Perform size selection using VAHTS DNA Clean Beads according to manufacturer's instruction.

### Next-generation sequencing

All sequencing experiments were performed with Illumina NovaSeq 6000 System. The service for scRNA-seq was provided by HaploX genomics center, while for snATAC-seq by Genergy Bio. For scRNA-seq, paired-end 150 bp with i7 8 bp sequencing strategy was used, while for snATAC-seq, paired-end 150 bp with i7 8 bp and i5 16 bp sequencing strategy was applied.

### Computational methods

### CASB barcode analysis

For scRNA-seq, raw barcode library FASTQ files were converted to barcode UMI count matrix using custom script leveraging the pysam (Li *et al*, 2009) package (https://github.com/pysam-developers/pysam). This procedure was similar with a previous method (McGinnis *et al*, 2019). Briefly, raw FASTQ files were first parsed to use only the reads where the first 16 bases of R1 perfectly match any of the cell barcodes predefined by Cell Ranger. Then, reads where the 20–28 bases of R2 align with at most 1 mismatch to any predefined sample barcodes were used. Reads were grouped by cell barcodes and duplicated UMIs were identified as reads where 17–26 bases of R1 exactly matched.

In snATAC-seq, sample barcodes were in R2 reads and cell barcodes were in R1 reads. Reads with sample and cell barcodes were first extracted from raw FASTQ files of snATAC library using custom script to get the cell-by-sample count matrix.

Barcode raw count matrix was first "CLR" normalized; then, HTODemux method (Stoeckius *et al*, 2018) implemented in Seurat package was used to define "doublets", "singlets" and "negatives".

For CASB combinatorial barcodes, HTODemux was used on set A barcodes, set B barcodes, and split-pool barcodes, respectively. Then, the "doublets" between samples were defined if it is doublet for both set A and B. The "doublets" within sample were defined if it is singlet for both set A and B but doublet for split-pool barcodes.

The "negatives" were defined if it is negative for either set A or B. And the rest were defined as "singlets".

### scRNA-seq gene expression analysis

FASTQ files were processed using Cell Ranger (10× Genomics, v3.1.0). The reads were aligned to the concatenated hg38-mm10 or hg38 reference using STAR (Dobin *et al*, 2013). Cell-associated barcodes were defined by Cell Ranger. Gene expression UMI count matrix (h5 file) was obtained using Cell Ranger with default parameters.

After the pre-processing, RNA UMI count matrices were prepared for scRNA-seq analysis using the "Seurat" R package (Butler *et al*, 2018). Cells with no more than 4,000 reads or 2,000 expressed genes were removed. Outlier cells with elevated mitochondrial gene expression were visually defined and discarded. Ribosomal genes and mitochondrial genes were then filtered out.

"sctransform" R package (Hafemeister & Satija, 2019) was used to normalize the RNA UMI count data and find highly variable genes. These variable genes were then used during principal component analysis (PCA). Elbow plot was used to select the top principal components. Then, these principal components were used for dimensionality reduction with UMAP and unsupervised clustering with Louvain method. Differential gene expression analysis was performed using the "FindMarkers" function in Seurat with "MAST" method (Finak *et al*, 2015). To quantify the magnitude of perturbation induced by drug on gene expression, we compared the proportion of each cell's $k$ ($k = 9$) nearest neighbors in principal component space with the "knn.covertree" R package. The proportion was normalized by the cell numbers of different groups.

For the combinatorial indexing dataset with 16 cell lines from 7 species (human, mouse, rat, dog, fly, hamster, and monkey), gene annotation gtf file and genome reference fasta file for each species were downloaded from Ensembl (version 102). The gtf file was processed using mkgtf (--attribute = gene_biotype:protein_coding) as suggested by cellranger. Cellranger were used to build a customed STAR index with all 7 species genome and gtf as input (--memgb = 128 --limitSjdbInsertNsj 2000000). FASTQ files were processed using Cell Ranger (10X Genomics, v3.1.0). The subsequent analysis was similar as described above.

### snATAC-seq data analysis

For plate-based dataset, reads from each cell were processed independently. The reads were trimmed using fastp (Chen *et al*, 2018) v0.19.5 (-a CTGTCTCTTATA --detect_adapter_for_pe -w 6 -- length_required 20 -q 30). CASB were extracted and counted while the reads without CASB were aligned to hg38 reference by bowtie2 v2.3.4.3 (-X 2000) (Langmead & Salzberg, 2012). Reads mapped to mitochondria or with low mapping quality (MAPQ < 20) were removed. Sambamba v0.7.0 (Tarasov *et al*, 2015) were used to sort and remove duplicated reads. Macs2 (Zhang *et al*, 2008) were used to call peaks with all the 192 cell deduplicated and sorted bam file as input. The peaks were centered at the summit and extend 500 bp both sides as features. FeatureCounts (Liao *et al*, 2014) were used to count the number of read pairs from each cell that were aligned within each feature, resulting in a peak/feature-by-cell count matrix. This matrix was binarized and peaks occurred in only on cell was filtered. Cell with no more than 300 peaks was also filtered. The resulting matrix was normalized by term frequency-inverse

document frequency (TF-IDF). Latent semantic indexing analysis was performed as applying singular value decomposition (SVD) on the normalized count matrix. Only the $2^{nd}$-$50^{th}$ dimensions after the SVD were passed to UMAP for 2D visualization. The distribution of CASB barcode read number from individual cells in plate-based snATAC-seq was used to separate labeled from unlabeled cells (the cutoff was 20,000).

For 10X Genomics dataset, after filtering out the reads with sample barcodes, Cellranger-atac (version 1.2.0) was used to process the raw FASTQ files. The reads were aligned to hg38 reference using BWA-MEM (Li & Durbin, 2009). The filtered peak-by-cell matrix (h5 file) obtained after cellranger-atac processing was used in the subsequent analysis. The matrix was first binarized. Cells of low quality (no more than 4,000 peaks or more than 500,000 peaks, percent of reads in peaks <= 30%, percent of peaks in ENCODE blacklist > 5%) were filtered out. Only cells defined by HTODemux as "singlet" were used for subsequent analysis. ATAC peaks with low coverage (< 50 cells) or ultra-high coverage (more than 2,000 cells) were also removed. The binarized count matrix was normalized by term frequency-inverse document frequency (TF-IDF). Latent semantic indexing analysis was performed as applying singular value decomposition (SVD) on the normalized count matrix. Only the $2^{nd}$-$50^{th}$ dimensions after the SVD were passed to UMAP for 2D visualization. Motif analysis was performed using chromVAR (Schep *et al*, 2017).

The predicted target genes of TF were defined by the nearest genes within 100 kb of the activated ATAC peaks with the TF motif at 12 h. The activated ATAC peaks were called by using FindMarkers function with parameter test.use="LR". The average expression levels of these predicted target genes were calculated using "AddModuleScore" function in Seurat package.

CoveragePlot in Signac (v0.2.5) (preprint: Stuart *et al*, 2020) package was used to plot the accessibility tracks for defined region.

### Gene function enrichment and network analysis using IPA software

The differentially expressed genes in cluster 1 and 2 compared to cluster 0 of MDA-MB-231 cells and genes differentially expressed upon compound treatment in cluster 0 cells were subjected to the Ingenuity Pathway Analysis (IPA) (QIAGEN) (Kramer *et al*, 2014) to gain insights into the gene functions. The "Diseases and Functions" module under the "Expression Analysis" was used for this purpose. In "Diseases and Functions" module, the analysis was restricted to "Molecular and Cellular Functions".

## Data availability

All next-generation sequencing data were submitted to Gene Expression Omnibus under the accession number GSE153116 (https://www.ncbi.nlm.nih.gov/geo/query/acc.cgi?acc=GSE153116). Scripts used for CASB barcode analysis is publicly available at https://github.com/GuipengLi/CASB.

**Expanded View** for this article is available online.

### Acknowledgements
This work was supported by Shenzhen Key Laboratory of Gene Regulation and Systems Biology, Shenzhen-Hong Kong Institute of Brain Science-Shenzhen

Fundamental Research Institutions (Grant No. 2021SHIBS0002), Shenzhen Science and Technology Program (Grant No. KQTD20180411143432337 and JCYJ20190809154407564), and National Natural Science Foundation of China (Grant No. 31970601, 31701237 and 31900431). The authors acknowledge the Center for Computational Science and Engineering of SUSTech for the support on computational resource and acknowledge the SUSTech Core Research Facilities for technical support.

## Author contributions

WC and LF developed the concept of the project. LF, ZS, QZ, HC, YL, and JZ designed and performed experiments. GL performed bioinformatic analysis. WL and WW assisted in performing experiments. WC, LF, GL, and YH reviewed and discussed results. WC, LF, and GL wrote the manuscript.

## Conflict of interest

The authors declare that they have no conflict of interest.

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
