## [Review Process File · Molecular Systems Biology]

CASB: A concanavalin A-based sample barcoding strategy for single-cell sequencing

Liang Fang, Guipeng Li, Zhiyuan Sun, Qionghua Zhu, Huanhuan Cui, Yunfei Li, Jingwen Zhang, Weizheng Liang, Wencheng Wei, Yuhui Hu, and Wei Chen

DOI: [10.15252/msb.202010060](https://doi.org/10.15252/msb.202010060)

Corresponding author(s): Wei Chen (chenw@sustc.edu.cn) , Liang Fang (fangl@sustech.edu.cn)

Review Timeline:

Submission Date:	15th Oct 20
Editorial Decision:	19th Oct 20
Revision Received:	20th Oct 20
Editorial Decision:	18th Nov 20
Revision Received:	3rd Feb 21
Editorial Decision:	26th Feb 21
Revision Received:	7th Mar 21
Accepted:	9th Mar 21

Editor: Maria Polychronidou

Transaction Report:

Thank you for submitting your manuscript "A concanavalin A-based sample barcoding strategy for single-cell sequencing" to Molecular Systems Biology.

I have now read your manuscript and discussed it with the team, and we think that the presented approach seems interesting. I am glad to inform you that we have decided to send the manuscript out for review.

However, before we send the manuscript to potential referees, I think that some more direct comparisons of CASB to alternative existing approaches would need to be included. Even if this is only by discussion, we think that the manuscript would benefit from clearer descriptions of the advantages and superiority of CASB compared to specific alternative methods. This would be quite useful for the referees.

1st Revision - Editorial Decision**18th Nov 2020**

Thank you again for submitting your work to Molecular Systems Biology. We have now heard back from the three referees who agreed to evaluate your study. Overall, the reviewers acknowledge that the proposed method seems to be a relevant contribution to the single-cell biology field. However, they raise a series of concerns, which we would ask you to address in a major revision.

Without repeating all the points listed below, some of the more fundamental issues are the following:

- The reviewers point out that a direct and systematic comparison to other methods is lacking. Such a comparison needs to be included and the advantages of CASB need to be clearly demonstrated. As suggested by the reviewers, a table comparing the different methods would indeed be helpful.
- The applications of CASB to combinatorial indexing and combined RNA-seq/ATAC-seq assays should ideally be demonstrated experimentally (as reviewer #1 recommends). This would add value to the study.
- Potential artifacts caused by ConA (e.g. aggregation or effects on the transcriptome), should be examined in further detail. Related to this, reviewer #2 also points out the high rate of doublets in some of the experiments, which needs to be carefully assessed.
- Reviewers #1 and #2 recommend expanding and clarifying the results of the specific application of CASB to examine drug responses, to make a stronger case for the success of the method.
- Reviewer #3 raises an important point (point #1) related to the potential bias of the method towards certain cell subpopulations in a mixture.
- In line with point #4 of reviewer #3 we would ask you to make sure that the code is made available. The methodology should be described in detail, and the manuscript would benefit from including protocols using our Structured Methods format.

Reviewer #1:

Here the authors describe a Concanavalin A based sample barcoding strategy (CASB) to label samples. As a proof of concept, they used CASB to label cells with different treatments and show it can be used in both scRNA-seq and scATAC-seq. Together with other recently described methods including MULTI-seq, sci-PLEX and ClickTags. CASB provides an alternative way of sample indexing. However, we are not convinced that CASB is comparable or better than current methods with current data. Here are a few major concerns that we would like the authors to address in any revision:

A more systematic discussion/comparison with other techniques. The authors mentioned other techniques for sample multiplexing have limitations including scalability, universality and potential to introduce artifacts. Can the authors provide a table or extended discussion on both the disadvantages and advantages of each method? From our perspective, compared to CASB, MULTI-seq, sci-PLEX and ClickTags all provide more highly scalable methods to label samples. It's therefore unfair to focus on CITE-seq as a comparator for scalability as the authors do in the discussion. On the other hand, one potential advantage of CASB over other techniques is that it can be used in scATAC-seq and it is simpler than many existing strategies for multiplexing single-cell samples. The strengths and limitations of CASB treatment should be thoroughly discussed and compared fairly with existing methods.

Combinatorial indexing is a key strategy in sample multiplexing. This will likely work well for CASB, but it is not demonstrated in the manuscript. Similar claims about RNA/ATAC co-assays, while highly plausible, were also not demonstrated. Ideally, the authors would demonstrate this. If this is not possible, they should make it clear that these could work in principle but have not yet been demonstrated.

ConA treatment could introduce transcriptional artifacts and agglutination (aggregation) of cells. The data in figure 1EV1-H are convincing for this cell type, but users will want to be sure that ConA treatment does not induce transcriptional artifacts in their cell type of interest. Because this concern is in line with the known biological activity of ConA and other lectins, how could potential users know that CASB will not generate artifacts?

In the scATAC-seq experiment, a ~10% doublet rate (305/2890 cells) was observed at low nuclei loading rate. Is this due to agglutination of nuclei prior to sample loading? Images of single-cell and single-nuclei populations before and after ConA treatment would be one way to demonstrate that ConA does not cause cells or nuclei to aggregate.

More thorough analyses of biological experiments. Gene differential expression analysis and an investigation of the transcriptional effects of the perturbations shown in Figure 2E-G would improve the manuscript and demonstrate the success of the method. We don't have a sense of the extent of perturbation performed.

Minor issues:

It will be really helpful if the authors can show the signaling pathway of INF- γ in Figure 3

Line 34,35: When discussing combinatorial indexing, papers including Cusanovich, Darren A., et al. (2015), Cao, Junyue, et al. (2017), Cao, Junyue, et al. (2018), Cao, Junyue, et al.(2019) should be cited.

Line 60, own limitations instead of own liabilities?

Reviewer #2:

Fang et al. utilized biotinylated Concanavalin A to label samples with specific DNA barcodes. They demonstrated that this method could be applied to both single-cell RNA-seq and ATAC-seq pipelines. Using this sample multiplexing strategy, they revealed the transcriptomic dynamics and chromatin accessibility changes in the cell lines treated with multiple drugs and cytokines. While CASB could be a useful universal tool to the field, I have several concerns where more clarity is needed:

1.As this method is used for sample multiplexing, rigorous assessment of cell doublet rate is essential to prove the cleanness of the assay and to reveal any potential issues. However, except in the first cell-line-mixture experiment (page 6, the last paragraph), where the high doublet rate ($3962/12068 \times 2$) was likely caused by cell overloading (~12k cells input), no explanation is provided to justify the high doublet rates in other experiments ($305/2890 \times 2$ in Fig. 3B and $294/3407 \times 2$ in Fig. EV4B), where a typical 2~3% doublet rate should be expected with 3k cells input.

2.Regarding the three clusters identified on MDA-MB-231 (Fig. 2E), what are the markers of each of the clusters? Is there any experimental evidence (e.g., double immunostaining) to support the clustering result? There is also evidence of sub-cluster within cluster 1. How was the clustering resolution determined?

3.In Fig 4D, it's not clear what each point represents just from the figure legend. But it seems that, despite being lower than their counterparts in cluster 2, the expression of NF- κ B target genes in cluster 0 at later time points (8h and 12 h) is actually higher than that in cluster 2 at earlier time points (4h and 6h). It contradicts the pattern of CXCL10 shown in Fig. 4E, which only expressed in cluster 2, no matter the time points, but not in any of the cells in cluster 0. It's also different from Fig. 3E, where the highest NF- κ B regulatory activities are mostly evidenced in the early (0h) and the late (12h) time points.

Additional points:

1.Could the authors explain why nuclei have higher tagging efficiency than cells (120,000 in Fig. EV1A vs. 50,000 in Fig. 1B), given much smaller membrane area?

2.UMAP with UMI counts per cell for Fig 2G would help rule out sequencing depth, or low cell quality (after drug treatment) caused effects.

3.For sample indexing in the snATAC-seq experiment, correlation analysis should be added to compare the data quality generated from untreated and indexed samples.

4.A table to summarize the advantages (UMI capture efficiency, cost, etc.) of the CASB method over other platforms will help the readers to get its potential.

Reviewer #3:

Fang et al. introduce a novel tagging strategy for multiplexed single-cell analysis. Several strategies for tagging cells with barcoded oligos have been developed in the past years, including tagging of fixed cells with a click reaction (clickTag), antibody-based cell hashing, and lipid-based anchoring of barcodes (multi-seq). The goal of these strategies is to allow overloading of single-cell platforms such as 10x genomics (reducing costs) and, most importantly, to allow the simultaneous processing of samples in a way that minimizes/eliminates batch effects (e.g. different treatments or developmental stages). The authors' strategy radically differs from all previous methods, using concanavalin A-biotin to bind glycoproteins in the cell surface. The concanavalinA-biotin is assembled first into a labeling complex with streptavidin and a biotinylated oligo containing the sample barcode, then this complex is briefly incubated with cells or nuclei. This design confers great flexibility to the system and allows very rapid barcoding of different types of samples. Most importantly, the method allows barcoding not only for scRNA-seq but also scATAC-seq, being the first such strategies amenable to this application. The authors apply their method first to a breast cancer cell line to simultaneously profile scRNA-seq from 5 different drug perturbations assays. Cross-species labeling assays demonstrate the doublet-detection capacity of their methods, as well as the signal scalability of the tagging strategy and the lack of strong batch effects associated with this labeling. Overall, this novel method represent an important development for single-cell biology and it makes this work a strong candidate for publication.

I have one major concern and a few additional comments:

1. My main concern is about performance of this tagging strategy in different species and, most importantly, in different cell types within the same sample. Some smaller cells or cells with different glycoprotein surface composition may systematically fail to incorporate enough concanavalin-based tags, and therefore become literally invisible in when overloading single-cells (de facto inadvertently mixing their RNA/accessibility signal with other cells). The authors must systematically address this by using different species and cell populations. Of course, this does not need to be measured through repeated time-consuming and expensive single-cell RNA-seq/ATAC-seq experiments, but simply assaying barcode detection by qPCR.
2. The authors must report the number of reads employed in each experiment and, specially, which fraction of reads correspond to barcodes versus transcripts/tagmented DNA. The lack of control over barcode sequencing (since no splitting of the libraries is performed) may results in an extremely cost-ineffective experiment, so these statistics are necessary to evaluate this possibility.
3. The way scATAC data is analysed and presented is a bit unusual. For example, the authors should show the accessibility tracks for the different clusters/cell populations and additional QC stats should be reported.
4. Before publication it is essential that the authors release all the code used in the analyses, as well as any dedicated scripts. Similarly, a detailed protocol must be included in order to ensure the broad applicability of the method.
5. Although not exactly a tagging strategy, the authors should acknowledge and discuss that multiplexing scATAC methods exist (e.g. based on barcoded Tn5 tagmentation or barcode ligation

after tagmentation).

Reviewer #1 (Comments to the Author):

Here the authors describe a Concanavalin A based sample barcoding strategy (CASB) to label samples. As a proof of concept, they used CASB to label cells with different treatments and show it can be used in both scRNA-seq and scATAC-seq. Together with other recently described methods including MULTI-seq, sci-PLEX and ClickTags. CASB provides an alternative way of sample indexing. However, we are not convinced that CASB is comparable or better than current methods with current data.

Here are a few major concerns that we would like the authors to address in any revision:

A more systematic discussion/comparison with other techniques. The authors mentioned other techniques for sample multiplexing have limitations including scalability, universality and potential to introduce artifacts. Can the authors provide a table or extended discussion on both the disadvantages and advantages of each method? From our perspective, compared to CASB, MULTI-seq, sci-PLEX and ClickTags all provide more highly scalable methods to label samples. It's therefore unfair to focus on CITE-seq as a comparator for scalability as the authors do in the discussion. On the other hand, one potential advantage of CASB over other techniques is that it can be used in scATAC-seq and it is simpler than many existing strategies for multiplexing single-cell samples. The strengths and limitations of CASB treatment should be thoroughly discussed and compared fairly with existing methods. **Answer:**

Thanks for the suggestion. We now provided a table (Table EV1) to list the advantages and limitations of each method.

Indeed, it is unfair to focus on CITE-seq as a comparator for scalability, therefore, we removed the sentence. (Line 310-315)

Combinatorial indexing is a key strategy in sample multiplexing. This will likely work well for CASB, but it is not demonstrated in the manuscript. Similar claims about RNA /ATAC co-assays, while highly plausible, were also not demonstrated. Ideally, the authors would demonstrate this. If this is not

possible, they should make it clear that these could work in principle but have not yet been demonstrated.

Answer:

Indeed, we have been working on combinatorial indexing using CASB. We now included the data of such an experiment, where we applied two strategies, i.e., combinatorial barcoding and split-pool barcoding. In this experiment, a four-by-four combinatorial barcoding allowed 16 cell types from seven different species to be indexed, whereas another round of split-pool indexing with four additional barcodes increased the number of cell indexes to 64. As shown in Figure 5, 64 indexes with different barcode combinations allowed to assign cell origin as well as to further increase the efficiency in doublet detection in the scRNA-seq experiment. For the detailed results, please refer to line 271-302 and Figure 5 and EV7.

We do not have the data to demonstrate the compatibility of CASB with RNA/ATAC co-assays, but, given the similar workflow between the RNA/ATAC co-assay and snATAC-seq provided by 10X Genomics, we think it should work in principle. Nevertheless, we rephrased our text as suggested by the reviewer (Line 333-336).

ConA treatment could introduce transcriptional artifacts and agglutination (aggregation) of cells. The data in figure 1EV1-H are convincing for this cell type, but users will want to be sure that ConA treatment does not induce transcriptional artifacts in their cell type of interest. Because this concern is in line with the known biological activity of ConA and other lectins, how could potential users know that CASB will not generate artifacts?

Answer:

To investigate the potential effect of CASB labelling on cell transcriptome in other cell types, in the combinatorial indexing experiment, we indexed eight cell types from human (HEK-293T, RPE-1, Jurkat, K-562, HCT116, HepG2, HeLa and 786-0), four from mouse (RAW264.7, CT26 and 4T1), and one each from rat (REF), dog (MDCK), hamster (CHO), monkey (Vero) and drosophila (S2) using CASB technique. Meanwhile, before loading on 10X system, equal number of unlabeled cells from each cell type were pooled together with the labeled cells. Notably, given ConA is a T cell mitogen, we intentionally included 'Jurkat', a T cell leukemia cell line. As shown in Figure 5H and EV7F, comparing labeled and unlabeled cells, for all the 16 cell types, single cell profiles were intermingled and cumulative transcriptome profile showed good correlation, suggesting CASB did not induce transcriptional artifacts. The reason why CASB did not affect transcriptome is likely because all the processing steps are performed on ice, where cells are kept in a quiescent state and unable to respond to any stimuli. For the detailed results, please see line 271-302 and Figure 5 and EV7.

In the scATAC-seq experiment, a ~10% doublet rate (305/2890 cells) was observed at low nuclei loading rate. Is this due to agglutination of nuclei prior to sample loading? Images of single-cell and single-nuclei populations before and after ConA treatment would be one way to demonstrate that ConA does not cause cells or nuclei to aggregate.

Answer:

To investigate whether ConA could cause cells or nuclei to aggregate, we took images of single-cell and single-nuclei populations with or without ConA treatment, as suggested by the reviewer and also quantified the singlet rate by flow cytometry. As shown in Fig EV1B&C, ConA did not induce cell or nucleus aggregation.

The unexpected high doublet rate is likely due to suboptimal condition of our 10X genomics equipment in the particular period, when these two experiments were performed. In the latest combinatorial indexing experiment, where we multiplexed 16 different cell lines and loaded the droplet system with standard number of cells (about 8000 cells were captured with sufficient reads), we got a doublet rate of 12%, which is within expected range. As suggested in user guide of 10X genomics, when 8000 cells are recovered, the doublet rate is around 6.1%. Since this doublet rate is calculated by mixing equal number of human and mouse cells, the true doublet rate should be 2x 6.1%, which is very close to the doublet rate in this experiment.

More thorough analyses of biological experiments. Gene differential expression analysis and an investigation of the transcriptional effects of the perturbations shown in Figure 2E-G would improve the manuscript and demonstrate the success of the method. We don't have a sense of the extent of perturbation performed.

Answer:

We have performed gene differential expression analysis for sensitive cell population after drug perturbation, which revealed that OSI-027, Niraparib and Rucaparib induced expression alteration of 613, 365 and 296 genes ($|\log_{2}FC| > 0.25$, P-value < 0.05 , Dataset EV1) in the sensitive cell population, respectively, which are highly enriched in cell death and survival pathway (Fig EV3F), again demonstrating the sensitivity of this cell population to these treatments.

Minor issues:

It will be really helpful if the authors can show the signaling pathway of INF- γ in Figure 3.

Answer:

Thanks for the suggestion. We have included a scheme of INF- γ pathway as Figure 3B.

Line 34,35: When discussing combinatorial indexing, papers including Cusanovich, Darren A., et al. (2015), Cao, Junyue, et al. (2017), Cao, Junyue, et al. (2018), Cao, Junyue, et al. (2019) should be cited.

Answer:

We have cited these important articles in our manuscript. (Line 41-42)

Line 60, own limitations instead of own liabilities?

Answer:

We have modified accordingly. (Line 66)

Reviewer #2 (Comments to the Author):

Fang et al. utilized biotinylated Concanavalin A to label samples with specific DNA barcodes. They demonstrated that this method could be applied to both single-cell RNA-seq and ATAC-seq pipelines. Using this sample multiplexing strategy, they revealed the transcriptomic dynamics and chromatin accessibility changes in the cell lines treated with multiple drugs and cytokines. While CASB could be a useful universal tool to the field, I have several concerns where more clarity is needed:

1. As this method is used for sample multiplexing, rigorous assessment of cell doublet rate is essential to prove the cleanness of the assay and to reveal any potential issues. However, except in the first cell-line-mixture experiment (page 6, the last paragraph), where the high doublet rate ($3962/12068 \times 2$) was likely caused by cell overloading (~12k cells input), no explanation is provided to justify the high doublet rates in other experiments ($305/2890 \times 2$ in Fig. 3B and $294/3407 \times 2$ in Fig. EV4B), where a typical 2~3% doublet rate should be expected with 3k cells input.

Answer:

The unexpected high doublet rate is likely due to suboptimal condition of our 10X genomics equipment in the particular period, when these two experiments were performed. In the latest combinatorial indexing experiment, where we multiplexed 16 different cell lines and loaded the droplet system with standard number of cells (about 8000 cells were captured with sufficient reads), we got a doublet rate of 12%, which is within expected range. As suggested in user guide of 10X genomics, when 8000 cells are recovered, the doublet rate is around 6.1%. Since this doublet rate is calculated by mixing equal number of human and mouse cells, the true doublet rate should be $2 \times 6.1\%$, which is very close to the doublet rate in this experiment.

To investigate whether ConA could cause cells or nuclei to aggregate, as suggested by the first reviewer, we took images of single-cell and single-nuclei populations with or without ConA

treatment and also quantified the singlet rate by flow cytometry. As shown in Fig EV1B&C, ConA did not induce cell or nucleus aggregation.

2. Regarding the three clusters identified on MDA-MB-231 (Fig. 2E), what are the markers of each of the clusters? Is there any experimental evidence (e.g., double immunostaining) to support the clustering result? There is also evidence of sub-cluster within cluster 1. How was the clustering resolution determined?

Answer:

Indeed, specific marker genes could be found for each cell cluster in scRNA-seq data, which we now added in the revised manuscript as Fig EV4. Given one of the key cellular programs differentially activated in cluster 0 vs. cluster1/2 cells is cellular movement, we focused on VIM and KRT18, two cytoskeleton proteins, for immunofluorescent analysis. As shown below, cells expressing only high level of VIM (cluster 1 cell) or KRT81 (cluster 2 cell) could be observed together with cells expressing both proteins at relatively low level (cluster 0 cell). Notably, MDA-MB-231 cells presented very heterogeneous shape in tissue culture, demonstrating again a high heterogeneity within the cell line.

Figure legend: Immunofluorescent staining of VIM (Green), KRT81 (Red) and cell nuclei (Blue). cells expressing only high level of VIM (cluster 1 cell) or KRT81 (cluster 2 cell) could be observed together with cells expressing both proteins at relatively low level (cluster 0 cell).

As suggested by the reviewer, a sub-cluster could indeed be observed within Cluster 1 in the UMAP projection (Fig 2E), but it is mainly due to the different level of total UMI count rather than the different transcriptome profile (shown below, also in Fig EV3B).

Figure legend: Transcriptome-based UMAP of untreated and OSI-027-treated MDA-MB-231 cells, in which UMI count of individual cells was indicated.

Given three main cell clusters could be visually identified consistently in UMAP projection containing only untreated cell or untreated cell together with OSI-027-, Niraparib- or Rucaparib-treated cells, we defined cell clusters with Seurat package and determine the resolution to fit the visual pattern of cell clusters.

3. In Fig 4D, it's not clear what each point represents just from the figure legend. But it seems that, despite being lower than their counterparts in cluster 2, the expression of NF- κ B target genes in cluster 0 at later time points (8h and 12 h) is actually higher than that in cluster 2 at earlier time points (4h and 6h). It contradicts the pattern of CXCL10 shown in Fig. 4E, which only expressed in cluster 2, no matter the time points, but not in any of the cells in cluster 0. It's also different from Fig. 3E, where the highest NF- κ B regulatory activities are mostly evidenced in the early (0h) and the late (12h) time points.

Answer:

In Fig 4D, each dot represents a cell, and Y-axis represents the average expression level of NF- κ B target genes. We now have clarified it in the figure legend. (Line 769-772)

Based on Fig 3E and F (now Fig 3F and G), as pointed out in the manuscript, the NF- κ B activity is not significantly changed across the different timepoints, instead is highly variable at all time points, likely due to the existence of two cell populations with low and high NF- κ B activity, respectively. To check whether such heterogeneous NF- κ B activity can have an effect on gene expression, we predicted the NF- κ B target genes based on the presence of active ATAC peaks containing NF- κ B motif in their vicinity (within 100 kb). Although this strategy was often used

to associate gene to ATAC peaks, we have to admit that such prediction would result in a list of target genes with many false positive as well as false negatives. Furthermore, as we know, the relationship between TF binding and target gene expression is often not straightforward, i.e., the expression of “target” genes may not solely depend on the TF binding. Given the two uncertainties, we could only draw any conclusions based on the global trend as shown in Figure 4D.

As suggested by the reviewer, some predicted NF- κ B target genes were also expressed in cluster 0 cells with low NF- κ B activity. These genes might be false positive predictions, or could be induced by other transcription factors induced by INF- γ . For example, among the 1030 predicted NF- κ B target genes, 297 and 234 of them were also predicted as IRF and STAT target genes, respectively. In contrast, well-known NF- κ B target genes, whose expression absolutely requires NF- κ B activity under INF- γ stimulation, such as CXCL10 and 11, the INF- γ treatment was unable to trigger the expression in cells with low NF- κ B activity.

Additional points:

1. Could the authors explain why nuclei have higher tagging efficiency than cells (120,000 in Fig. EV1A vs. 50,000 in Fig. 1B), given much smaller membrane area?

Answer:

Indeed, we were also surprised by this phenomenon. But, of note, we had not saturated cellular or nuclear surface with CASB barcodes in these experiments, thus we did not know whether the maximal tagging efficiency of the nuclear surface is also higher than that of the cellular surface. This result only suggested that, at the same concentration of CASB labeling complex, nuclei were more efficiently tagged.

To answer the question, we performed fluorescent labeling experiment in which we firstly saturated cellular or nuclear surface with excessive amount of biotinylated-ConA, and then labeled cells and nuclei with excessive amount of streptavidin-conjugated fluorophore. As revealed by flow cytometry quantification (Figure below), nucleus again demonstrated a much higher labeling efficiency than cell.

Figure legend: The same number of K562 cells and nuclei were firstly labeled with excessive amount of biotinylated ConA and washed, and then labeled with excessive amount of streptavidin-conjugated fluorophore. After another washing, all samples were analyzed by flow cytometry. The mean fluorescent intensity of each sample was plotted against the usage of biotinylated ConA. The concentration of biotinylated ConA used in this experiment is 48-120 times higher than in scRNA-seq experiment.

We speculate this could potentially be due to the more complex environment on the cellular membrane, where other large molecules may hinder the binding of CASB complex to the glycoprotein.

2. UMAP with UMI counts per cell for Fig 2G would help rule out sequencing depth, or low cell quality (after drug treatment) caused effects.

Answer:

Thanks for the suggestion. We have included UMAP with UMI counts of each cell as Fig EV3B, in which UMI count of individual cells was indicated and did not show cluster or treatment bias.

3. For sample indexing in the snATAC-seq experiment, correlation analysis should be added to compare the data quality generated from untreated and indexed samples.

Answer:

As suggested by the reviewer, we have performed such a control snATAC-seq experiment. In this experiment, CASB labeled HAP1 cells were pooled with the same number of unlabeled HAP1 cells and subjected for bulk tagmentation, FACS sorted into two 96-well plates and subsequent ATAC-seq library preparation. In total, 184 out of 192 cells were obtained with sufficient reads (Fig EV5B). We then separated CASB labeled HAP1 cells from unlabeled cells based on the number of CASB barcode reads in individual cells (Fig EV5B). CASB labeled and unlabeled cells were intermingled in UMAP projection according to the ATAC signal (Fig EV5C), suggesting no influence of CASB labeling on epigenomic profile. This was further

confirmed, when the cumulative ATAC signal of the labeled and unlabeled cells was compared: the correlation between the labeled and unlabeled cells was similar as that between the two plates (Fig EV5D). (Line 213-223)

4. A table to summarize the advantages (UMI capture efficiency, cost, etc.) of the CASB method over other platforms will help the readers to get its potential.

Answer:

Thanks for the suggestion. We now provided a table as Table EV1 to list the advantages and limitations of each method.

Reviewer #3 (Comments to the Author):

Fang et al. introduce a novel tagging strategy for multiplexed single-cell analysis. Several strategies for tagging cells with barcoded oligos have been developed in the past years, including tagging of fixed cells with a click reaction (clickTag), antibody-based cell hashing, and lipid-based anchoring of barcodes (multi-seq). The goal of these strategies is to allow overloading of single-cell platforms such as 10x genomics (reducing costs) and, most importantly, to allow the simultaneous processing of samples in a way that minimizes/eliminates batch effects (e.g. different treatments or developmental stages). The authors' strategy radically differs from all previous methods, using concanavalin A-biotin to bind glycoproteins in the cell surface. The concanavalin A-biotin is assembled first into a labeling complex with streptavidin and a biotinylated oligo containing the sample barcode, then this complex is briefly incubated with cells or nuclei. This design confers great flexibility to the system and allows very rapid barcoding of different types of samples. Most importantly, the method allows barcoding not only for scRNA-seq but also scATAC-seq, being the first such strategies amenable to this application. The authors apply their method first to a breast cancer cell line to simultaneously profile scRNA-seq from 5 different drug perturbations assays. Cross-species labeling assays demonstrate the doublet-detection capacity of their methods, as well as the signal scalability of the tagging strategy and the lack of strong batch effects associated with this labeling. Overall, this novel method represents an important development for single-cell biology and it makes this work a strong candidate for publication.

I have one major concern and a few additional comments:

1. My main concern is about performance of this tagging strategy in different species and, most importantly, in different cell types within the same sample. Some smaller cells or cells with different glycoprotein surface composition may systematically fail to incorporate enough concanavalin-based tags, and therefore become literally invisible in when overloading single-cells (de facto inadvertently mixing their RNA/accessibility signal with other cells). The authors must systematically address this by

using different species and cell populations. Of course, this does not need to be measured through repeated time-consuming and expensive single-cell RNA-seq/ATAC-seq experiments, but simply assaying barcode detection by qPCR.

Answer:

To demonstrate its compatibility with different cell types and from different species, in the new combinatorial indexing experiment, we multiplexed eight cell types from human (HEK-293T, RPE-1, Jurkat, K-562, HCT116, HepG2, HeLa and 786-0), four from mouse (RAW264.7, CT26 and 4T1), and one each from rat (REF), dog (MDCK), hamster (CHO), monkey (Vero) and drosophila (S2) using CASB technique. Meanwhile, to reveal the negligible influence of CASB on cell transcriptome, before loading on the 10X system, equal number of unlabeled cells from each cell type were pooled. As shown in Fig EV7C, the cell from the 16 cell lines were all labeled with sufficient amount of total CASB barcode with limited variability in labeling efficiency among different cell lines, demonstrating the universality of CASB technique. As shown in Figure 5, 64 different barcode combinations allowed both to assign the cell origin and to improve the efficiency of doublet detection in scRNA-seq experiment. For the detailed results, please see line 271-302 and Figure 5 and EV7.

2. The authors must report the number of reads employed in each experiment and, specially, which fraction of reads correspond to barcodes versus transcripts/tagmented DNA. The lack of control over barcode sequencing (since no splitting of the libraries is performed) may results in an extremely cost-ineffective experiment, so these statistics are necessary to evaluate this possibility.

Answer:

Thanks for the suggestion. We have added this information in the manuscript. In the HAP1 snATAC-seq experiment, of a total of 345,538,181 sequencing reads, the 5,095,947 (about 1.5%) were derived from barcodes. (Line 233-234)

Actually, for scRNA-seq experiments, CASB barcode and transcriptome library were separated by size selection before next-generation sequencing library construction, which enables the sequencing of the two libraries separately at a user-defined depth.

3. The way scATAC data is analyzed and presented is a bit unusual. For example, the authors should show the accessibility tracks for the different clusters/cell populations and additional QC stats should be reported.

Answer:

We have added additional information in the manuscript. The number of ATAC peaks detected in individual cells was shown in Fig EV5F. The cumulative ATAC signal around CXCL10 and

11 genes in two cell clusters with different NF- κ B activity at different time points were demonstrated in Fig EV6D.

4. Before publication it is essential that the authors release all the code used in the analyses, as well as any dedicated scripts. Similarly, a detailed protocol must be included in order to ensure the broad applicability of the method.

Answer:

All next-generation sequencing data were submitted to GEO under the accession number GSE153116. Scripts used for CASB barcode analysis is publicly available at <https://github.com/GuipengLi/CASB>. The detailed protocol was also included in the Method section in 'Structured Methods' format.

5. Although not exactly a tagging strategy, the authors should acknowledge and discuss that multiplexing scATAC methods exist (e.g. based on barcoded Tn5 tagmentation or barcode ligation after tagmentation).

Answer:

Thanks for pointing it out. We have cited the relevant references in the new version of the manuscript. (Line 41-42)

Thank you for sending us your revised manuscript. We have now heard back from the three reviewers who were asked to evaluate your study. Overall, the reviewers appreciate the thorough response to their concerns and think that the study has improved as a result of the performed revisions. However, as you will see below, reviewers #1 and #2 still list a few remaining concerns, which we would ask you to address in a revision. Most of them can be addressed by providing clarifications and/or discussions.

When you revise your manuscript, we would also ask you to address the following pending editorial issues.

Reviewer #1:

The authors performed a thorough response to reviewer comments and addressed our concerns from the first submission. The paper is now thorough and fairly complete. Overall, the method seems to perform quite well for RNA-seq and reasonably well for ATAC-seq. We only have one remaining comment related to a new section on combinatorial indexing.

The authors perform a 16-sample multiplexing experiment that uses 4x4 sample multiplexing. Then, they pool all samples, split the mixture into four groups, and add an additional third multiplexing oligo. It's good to see a demonstration of combinatorial multiplexing, but we are confused by the

pool-and-split step. Once the samples are pooled, they cannot be uniquely labeled, so the 3rd oligo does not add any sample identifying information. This is clear from Figure 5E. While this step does not invalidate the experiment, its purpose and value is unclear, and should be clarified.

Other than that, the authors performed well-controlled experiments, especially including unlabeled cells along with labeled cells in the same experiment, and they thoroughly explored the data.

Reviewer #2:

I commend the effort the authors have made to address those comments and I'm satisfied with most of their response. The inclusion of the application of CASB in combinatorial indexing is also encouraging and may open new avenues of scaling up single-cell experiments at low cost. However, I still have a few points of criticism on this revised manuscript.

1. My biggest concern is still about the high doublet rates described here, which are pretty unusual in single-cell analysis. Although the imaging study and flow cytometry suggested that cell/nuclei aggregation might not be a problem, it would be better if the authors can directly show that a low cell doublet rate can be achieved from a lower input (2000~5000 cell/nuclei) single-cell RNA/ATAC-seq data when applying CASB.

2. In Figure 5F, there are 4 clusters (in brown, in the left bottom part of the figure) that seem to come from the same cell line (HEK?). But in Figure 5C, they were separated by different combinatorial barcodes. Is this batch effect? In Figure 5G, K562 cells (in cyan, in the bottom part of the figure) were separated into two clusters. Can the authors discuss why?

3. The Pearson's correlation coefficient showed in Figure EV5D is very low. Is that due to the sparsity of ATAC-seq? Can the authors aggregate nearby peaks (over genomic bins) to mitigate the sparsity issue and do the plot again, and see if there is any improvement?

Reviewer #3:

The authors satisfactorily addressed my concerns about tagging efficiency in different cell types and species. Congratulations on an important innovation in the single-cell genomics field.

Point-by-point response to the referees' comments**General response to the reviewers:**

We thank again the three reviewers for their time and appreciate their comments. During this revision, we have carefully addressed questions from reviewers #1 and #2, as shown below marked in red.

Reviewer #1 (Comments to the Author):

The authors performed a thorough response to reviewer comments and addressed our concerns from the first submission. The paper is now thorough and fairly complete. Overall, the method seems to perform quite well for RNA-seq and reasonably well for ATAC-seq. We only have one remaining comment related to a new section on combinatorial indexing.

The authors perform a 16-sample multiplexing experiment that uses 4x4 sample multiplexing. Then, they pool all samples, split the mixture into four groups, and add an additional third multiplexing oligo. It's good to see a demonstration of combinatorial multiplexing, but we are confused by the pool-and-split step. Once the samples are pooled, they cannot be uniquely labeled, so the 3rd oligo does not add any sample identifying information. This is clear from Figure 5E. While this step does not invalidate the experiment, its purpose and value is unclear, and should be clarified.

Other than that, the authors performed well-controlled experiments, especially including unlabeled cells along with labeled cells in the same experiment, and they thoroughly explored the data.

Answer :

Thanks for the question. Indeed, the 3rd oligo from pool-and-split step did not uniquely label 16 different cell line, and, therefore, was not helpful on identifying cell types. However, it increased the barcode combinations from 16 to 64 for the whole cell pool, which could help to identify the cell doublets that were formed by the same cell type (referred to as 'doublets within sample'). In our experiment, the 3rd oligo helped to identify 26 doublets within sample (Fig. EV7A). Given the high multiplexity and standard loading rate of the droplet system in this particular experiment, we do not expect to observe a lot of cell doublets within sample. However, this strategy will help to efficiently identify cell doublets within sample when only a few samples are multiplexed in a superloading experiment. We have now clarified our intension of using the 3rd oligo in the manuscript (Line 280-283).

Reviewer #2 (Comments to the Author):

I commend the effort the authors have made to address those comments and I'm satisfied with most of their response. The inclusion of the application of CASB in combinatorial indexing is also encouraging and may open new avenues of scaling up single-cell experiments at low cost. However, I still have a few points of criticism on this revised manuscript.

1. My biggest concern is still about the high doublet rates described here, which are pretty unusual in single-cell analysis. Although the imaging study and flow cytometry suggested that cell/nuclei aggregation might not be a problem, it would be better if the authors can directly show that a low cell doublet rate can be achieved from a lower input (2000~5000 cell/nuclei) single-cell RNA/ATAC-seq data when applying CASB.

Answer:

In the latest combinatorial indexing experiment, where we loaded the droplet system with standard number of cells (about 8000 cells were captured with sufficient reads), we got a doublet rate of 12% (calculated according to CASB barcodes), which is within expected range. As suggested in user guide of 10X genomics, when 8000 cells are recovered, the doublet rate is around 6.1%. Since this doublet rate is calculated by mixing equal number of human and mouse cells, the true doublet rate should be 2x 6.1%, which is very close to the doublet rate we calculated according to CASB barcodes.

Since the cell doublet rate could be affected by many different factors, such as the loading rate, sample property, hands-on experience, technical and equipment fluctuation, to fairly test whether CASB increased cell doublet rate, the best way would be to compare the doublet rate of CASB labeled and unlabeled cell in a single run of single-cell experiment. Therefore, to further exclude the potential of CASB to cause higher cell doublet rate, we compared the doublet rate of CASB labeled and unlabeled cells based on our combinatorial indexing experiment, where 16 different cell lines were multiplexed. Here, we considered only cell doublets consisting of samples from different species, which could be easily detected based on the commonly used genomic mapping information. Then we counted the number of cell doublets formed by 'two labeled cells' or 'two unlabeled cells', respectively. As an example, there were 120 and 94 unlabeled and labeled human-mouse doublets, respectively. The doublet rates were 4.27% ($120/(1931+757+120)$) and 3.72% ($94/(1663+770+94)$)(Fig R1A,B). The doublet rates of human-rat, -monkey, -hamster, -dog and -fly doublets were then calculated in the same way. As shown in Fig R1C, the cell doublet rates of CASB labeled cells were comparable to those of unlabeled cells for these different kinds of doublets, demonstrating that CASB does not increase cell doublet rate.

Fig R1. Comparing the doublet rate of CASB labeled and unlabeled cells in the combinatorial indexing experiment. A&B, Scatter plot depicting the number of UMIs associated with transcripts mapped to human or mouse genome. Human-mouse cell doublets revealed by genomic information were marked in green. There were 120 unlabeled and 94 labeled human-mouse doublets, respectively, with doublet rate of 4.27% and 3.72%. C, The doublet rates of unlabeled and labeled human-rat, -monkey, -hamster, -dog and -fly doublets. The cell doublet rates of CASB labeled cells were comparable to those of unlabeled cells for these different kinds of doublets.

2. In Figure 5F, there are 4 clusters (in brown, in the left bottom part of the figure) that seem to come from the same cell line (HEK?). But in Figure 5C, they were separated by different combinatorial barcodes. Is this batch effect? In Figure 5G, K562 cells (in cyan, in the bottom part of the figure) were separated into two clusters. Can the authors discuss why?

Answer:

In Figure 5F, cells were colored according to the transcriptomic feature revealed by Louvain algorithm. The reason why they were all colored in brown is because they have similar gene expression profiling according to Louvain algorithm. However, the 4 clusters in brown were actually from 3 different cell lines (HEK-293T, HeLa and HepG2). To clarify it, we labeled the names of human cell lines in Figure 5G (shown below).

The reason why K562 cells were separated into two clusters is the same as for HeLa, HEK-293T, HCT116 and MDA-MB-231 cells: cellular heterogeneity within a same cell line. This has been reported by many different studies (the latest one, please refer to *Kinker, et al., Nature Genetics volume 52, pages1208–1218(2020)*).

3. The Pearson's correlation coefficient showed in Figure EV5D is very low. Is that due to the sparsity of ATAC-seq? Can the authors aggregate nearby peaks (over genomic bins) to mitigate the sparsity issue and do the plot again, and see if there is any improvement?

Answer:

Indeed, it is due to the sparsity of snATAC-seq from only less than 100 cells. According to the suggestion, we now have aggregated 3, 5, 7 or 9 nearby peaks and performed the correlation analysis, respectively. As expected, the Pearson's correlation coefficients between labeled and unlabeled cells were increased from originally 0.276, to 0.435, 0.500, 0.535, 0.556, respectively, while those between two plates were increased from originally 0.143 to 0.290, 0.354, 0.389, to 0.411, respectively.

Fig R2. Scatterplot demonstrating the correlated epigenomic profiles between labeled and unlabeled as well as those between cells collected in plate 1 and 2. Due to the sparsity of snATAC-seq, 9 nearby peaks were aggregated to perform the correlation analysis. The correlation between the labeled and unlabeled cells was similar as that between the two plates. "R" means Pearson's correlation coefficient. Each dot presents the accumulated intensity of 9 nearby ATAC peak.

Reviewer #3 (Comments to the Author):

The authors satisfactorily addressed my concerns about tagging efficiency in different cell types and species. Congratulations on an important innovation in the single-cell genomics field.

Thank you again for sending us your revised manuscript. We are now satisfied with the modifications made and I am pleased to inform you that your paper has been accepted for publication.

Corresponding Author Name: Wei Chen

Journal Submitted to: MSB

Manuscript Number: MSB-2020-10060R